# Adaptations in wing morphology rather than wingbeat kinematics enable flight in small hoverfly species

Camille Le Roy[1]*, Nina Tervelde[1], Thomas Engels[2], Florian T Muijres[1]

[1]Experimental Zoology Group, Wageningen University & Research, Wageningen, Netherlands; [2]Aix-Marseille University, Marseille, France

## eLife Assessment

This **important** study addresses how wing morphology and kinematics change across hoverflies of different body sizes. The authors provide **convincing** evidence that there is no significant correlation between body size and wing kinematics across 28 species and instead argue that non-trivial changes in wing size and shape evolved to support flight across the size range. Overall, this paper illustrates the power and beauty of an integrative approach to animal biomechanics and will be of broad interest to biologists, physicists and engineers.

**Abstract** Due to physical scaling laws, size greatly affects animal locomotor ability and performance. Whether morphological and kinematic traits always jointly respond to size variation remains poorly known. Here, we examine the relative importance of morphological and kinematic changes in mitigating the consequence of size reduction on aerodynamic force production for weight support, focusing on the flight of hoverflies (Syrphidae). We compared the morphology of 28 hoverfly species, and the flight biomechanics and aerodynamics of eight species with body masses ranging from 5 to 100 mg. Our study reveals no significant effect of body mass on wingbeat kinematics among species, suggesting that morphological rather than kinematics changes compensate for the reduction in weight support associated with an isometric reduction in wing size. Computational fluid dynamics simulations confirmed that adaptations in wing morphology drive the ability of small hoverfly species to generate weight support, although variations in wingbeat kinematics among species cannot be entirely ignored. We show that smaller hoverflies have evolved relatively larger wings and aerodynamically more effective wing shapes, mitigating the reduction in aerodynamic weight support associated with isometric size reduction. Altogether, these results suggest that hoverfly flight underpins highly specialised wingbeat kinematics, largely conserved throughout evolution; instead, evolutionary adaptations in wing morphology enabled flight of small hoverflies.

**\*For correspondence:**
camille.leroy@wur.nl

**Competing interest:** The authors declare that no competing interests exist.

## Introduction

Evolutionary changes in size are common over the course of animal diversification (**Clauset and Erwin, 2008**; **Hanken and Wake, 1993**), strongly impacting many aspects of morphology, physiology, and behaviour. Investigating how changes in body size affect other traits (i.e. scaling) is thus essential to understand the causes of trait variation in animals. Traditionally rooted in the field of morphology and physiology (**Gould, 1966**; **Hill, 1950**; **Pélabon et al., 2014**), allometric studies have broadened to explore how more complex traits scale with animal size. These include locomotor kinematics (e.g. flight, stride, or swimming kinematics, **Bejan and Marden, 2006**; **Riskin et al., 2010**) and behaviour (e.g. mating or escape tactics, **Dial et al., 2008**). Although behavioural or biomechanical traits are

often more noisy and challenging to measure, their inclusion in allometric studies is crucial to better understand trait evolution (**Cloyed et al., 2021**). Indeed, most phenotypes involve both morphological and behavioural traits, each responding in potentially different ways to size variation (**Clemente and Dick, 2023**). Differences in scaling patterns between body size and other traits is generally most apparent between species and can be partly explained by neutral divergence during phylogenetic history (**Blomberg et al., 2003**). Understanding the origin of trait variation among species thus requires controlling for the influence of phylogeny on allometric relationships (**Labonte et al., 2016**; **Peattie and Full, 2007**).

In this study, we address the consequences of size variation between species (i.e. evolutionary allometry) and specifically question the relative importance of wing morphology versus wingbeat kinematic changes in mitigating the effect of size on flight ability.

Flight typically combines a suite of morphological and biomechanical traits, providing a relevant context to jointly examine how morphology and locomotor kinematics change with size. Moreover, the physical scaling laws governing flight allow drawing predictions on the scaling of morphology and kinematics in flying animals. These physical laws set both upper and lower bounds on the size of flying animals, resulting from the need to maintain weight support by producing upward-directed aerodynamic lift forces. In this context, there are two primary limitations for producing these forces: Limitations on (1) the muscular flight motor system and on (2) the flapping-wing-based propulsion system.

1. Just like terrestrial animals, larger flying animals face increasing constraints on muscle force production as they grow. This limitation arises because muscle force output scales with muscle cross-sectional area, and therefore under geometric similarity, it increases at a lower rate than body mass. Consequently, as body size increases, producing the muscle forces for maintaining weight support may require a positive allometry of muscle cross-sectional area (**Alexander, 2013**) however, alternative strategies such as changes in musculoskeletal gearing or adjustments in flight kinematics and behaviour can also help offset these scaling constraints (**Azizi et al., 2008**; **Norberg, 2007**). Conversely, smaller animals are not constrained by muscle force production but instead benefit from a favourable surface-to-volume ratio.

2. Next to this constraint on the muscular motor system, flying animals also face constraints stemming from the scaling between their flapping-wing-based propulsion system and body mass.

    a. For large flying animals, the aerodynamic lift ($L$) for weight support scales linearly with the surface area of the wing ($S$) and quadratically with the flight speed ($U_\infty$) as $L \sim S\, U_\infty^2$ (**Norberg, 2007**). Under geometric similarity, wing surface area scales with body mass ($m$) as $S \sim m^{2/3}$ (**Schmidt-Nielsen, 1975**). Thus, to maintain weight support during flight ($L = mg$, where $g$ is gravitational acceleration), larger flying animals exhibit disproportionately larger wings and tend to fly faster (**Norberg, 2012**). This positive allometry in wing surface area and flight speed with body mass is needed to generate enough lift to stay in the air.

    b. Smaller flying animals such as insects generally fly slower, and thus air movements over the wing are governed by wing flapping and less so by their forward movement. In the limiting case of hovering flight ($U_\infty = 0$ m s$^{-1}$), the airflow is generated entirely by the flapping wings, and the animal relies solely on this flapping-based propulsion to produce an upward aerodynamic force that balances its weight. This aerodynamic lift force produced by flapping wings scales linearly with the wing's second-moment-of-area ($S_2$) and quadratically with angular wing speed ($\omega$) as $L \sim S_2 \omega^2$ (**Weis-Fogh, 1973**). Thus, for small slow-flying animals, and particularly during hovering flight, the lift forces required for weight support scale primarily with the wing's second-moment-of-area, relative to the wing hinge (**Muijres et al., 2017**). For geometric similarity, this second-moment-of-wing-area scales with body mass as $S_2 \sim m^{4/3}$. Because this scaling factor of 4/3 is larger than one, a wing's second-moment-of-area decreases at a higher rate than body mass, and therefore wingbeat-induced aerodynamic lift reduces more rapidly than body weight. Thus, to maintain weight support ($mg = L \sim S_2\, \omega^2$), smaller flying animals require disproportionately larger wings, and/or they need to beat their wings at higher wingbeat frequencies or amplitudes, resulting in higher angular speeds of the beating wing. Notably, decreasing size also leads to a drop in Reynolds number, increasing the relative influence of viscous over inertial forces. Consequently, at the low Reynolds numbers typical of small insects, viscous forces may strongly affect both lift and drag forces produced during flapping flight (**Sane, 2003**).

In this study, we aim to test how physical scaling laws cause constraints on the flapping-wing-based propulsion system in small insects, and how this drives evolutionary adaptations in wing morphology and wingbeat kinematics. Here, we hypothesise that smaller insects have evolved

disproportionately larger wings and beat their wings at higher wingbeat frequencies, allowing them to maintain weight support despite the detrimental scaling effects of size reduction on their propulsion system.

The assumed increase in wingbeat frequency as size decreases is supported by the consistent negative relationship between wingbeat frequency and size observed in most flying animals (*Norberg, 2007*; *Rayner, 1988*; *Riskin et al., 2010*; *Tercel et al., 2018*). A general trend towards disproportionately larger wings in smaller species is, however, less striking among species (but see *Danforth, 1989*; *Ellington, 1984a*; *Outomuro et al., 2013*), suggesting that morphological changes in the propulsion system in response to size variation are not always straightforward.

Although empirical data often align with expectations drawn from physical laws, the scaling of morphology and kinematics with size can sometimes be challenging to predict. Selective pressures associated with ecological specialisation may influence the evolution of morphology and kinematics and conflict with the constraints imposed by physical scaling laws. Each component may then adjust for the constraints imposed by weight support in varying degrees. For example, changes in morphology have been shown to predominantly mitigate the negative consequences of body size reduction: among hummingbirds, a disproportionate increase in wing area – but little change in wingbeat kinematics – enables weight support among species of different sizes (*Skandalis et al., 2017*). Similarly, a positive allometry in wing area, but no significant change in wingbeat frequency, was found in relation to body size among species of bees (*Duell et al., 2022*; *Grula et al., 2021*). Such discrepancies with the general scaling expectation may point to a particular selective regime maintaining specialised flight kinematics and ability.

Here, we focused on an ecologically specialised insect group, the hoverflies, to test if the consequences of size variation are mostly mitigated by changes in wingbeat kinematics or in wing morphology. Hoverflies (Diptera: Syrphidae) are among the most species-rich groups of nectar-foraging insects and exhibit striking variation in size, ranging from 0.5 mg to over 200 mg (*Katzourakis et al., 2001*). Despite these large size differences, all adult hoverflies show exceptional hovering abilities. This highly specialised flight behaviour may have been promoted because it plays an important role in mating behaviour: male hoverflies aggregate at specific locations such as under a tree and hover steadily, waiting for passing females to pursue (*Gilbert, 1984*; *Heinrich and Pantle, 1975*). Hovering abilities may thus be correlated with mating success (*Downes, 1969*; *Gilbert, 1984*). It may also have been favoured in the context of foraging and navigating around flowers, albeit hovering may not directly relate to feeding as most species land on flowers to feed (*Gilbert, 1981*). Selection acting on the flight of hoverflies may interfere with general scaling expectations between body mass, wing morphology, and wingbeat kinematics.

We investigate the evolutionary response of wing morphology and wingbeat kinematics to weight support constraints in hovering hoverflies, specifically for small species. We focus on hovering flight as it is well defined ($U_\infty$=0 m s$^{-1}$), and because it is a functionally and energetically demanding flight mode (*Ellington, 1991*). During such hovering, the aerodynamic forces produced by the beating-wing-based propulsion system scale negatively with mass. We therefore hypothesise that tiny hoverflies have evolved relatively large wings or high wingbeat frequencies to maintain weight support during hovering. To test whether the potential adaptations in morphology or kinematics are dominant, we furthermore define two alternative null hypotheses: (1) by assuming that selection for strong hovering performance is limiting kinematic flexibility, we hypothesise that adaptations in wing morphology prevail over adaptations in wingbeat kinematics. (2) Alternatively, if flight behaviour is evolutionarily more labile than morphology (*Blomberg et al., 2003*), we expect that adaptations in wingbeat kinematics prevail over wing morphological adaptations.

We test these two null hypotheses by examining the scaling relationships between wing morphology and body mass in 28 hoverfly species ranging from 3 to 132 mg, and conjointly assessing how wing morphology and wingbeat kinematics scale with body mass for eight species ranging from 5 to 100 mg. We then combined computational fluid dynamic (CFD) simulations with quasi-steady aerodynamic modelling to estimate aerodynamic forces produced in hovering flight. Based on this, we determined how the relative changes in wingbeat kinematics and wing morphology contribute to producing weight support during hovering flight of hoverflies.

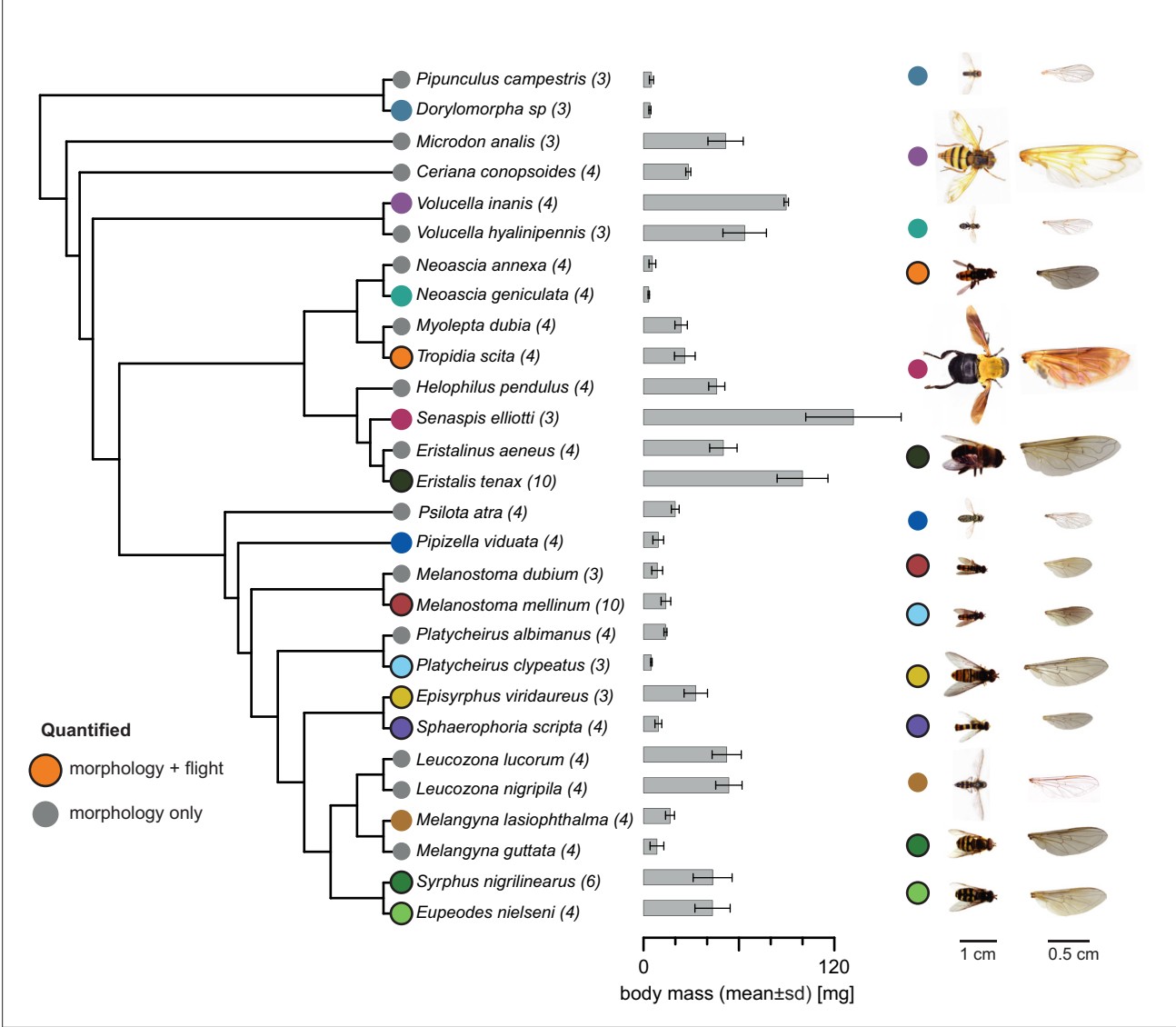

**Figure 1.** The studied hoverfly species shown with their phylogenetic relationships. Sample size as number of individuals for each species is in parentheses. Species marked with coloured circles are displayed on the right, and coloured circles with black outlines are of species for which both morphology and flight kinematics were quantified. Body mass is presented as mean values and standard deviation. Phylogeny was obtained from *Wong et al., 2023*.

The online version of this article includes the following figure supplement(s) for figure 1:

**Figure supplement 1.** The body mass of museum specimens was approximated using the relationship between thorax width and the fresh mass in wild-caught specimens.

## Results

In this study, we examined the morphology of 28 hoverfly species, with body masses ranging from 3 to 132 mg. Among them, we analysed the hovering flight biomechanics and aerodynamics of eight species, with body masses ranging from 5 to 100 mg. The sampled species were well spread throughout the phylogeny of hoverflies (*Figure 1*). Among the eight species for which we studied flight biomechanics, body mass variation was unevenly distributed. Only one species, *Eristalis tenax*, weighed more than 50 mg, while the two closely related species *Syrphus nigrilinearus* and *Eupeodes nielseni* had highly similar body masses (*Figure 1*).

Sexual dimorphism in morphological traits was observed in body mass, wing chord, and wing second-moment-of-area (*Supplementary file 1a*). However, the direction of this difference varied

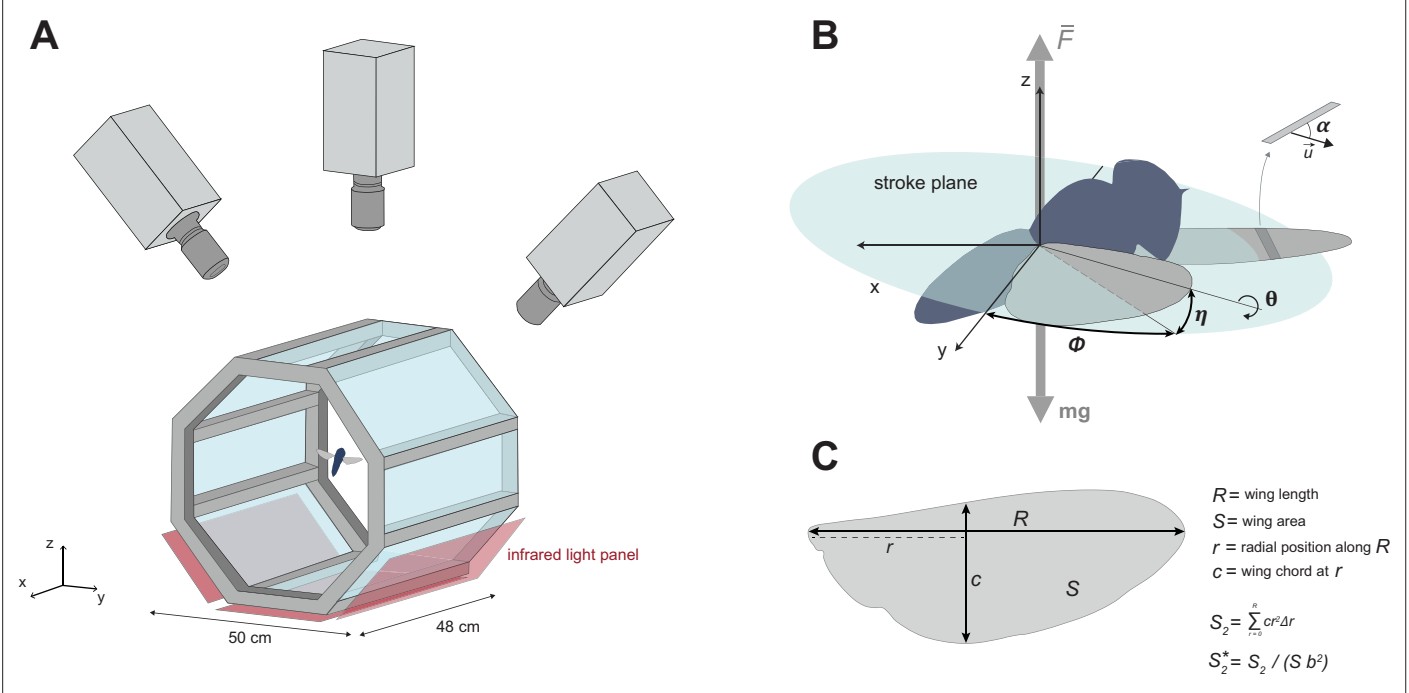

**Figure 2.** Quantifying the in-flight wingbeat kinematics and wing morphology of hoverflies. (**A**) Hoverflies were released in an octagon-shaped flight arena. We recorded stereoscopic high-speed videos of the flying hoverflies using three synchronised high-speed video cameras; from the videos, we reconstructed the three-dimensional body and wingbeat kinematics. Infrared light panels positioned as the bottom of the arena enabled high contrast between the flying insect and the background. (**B**) Conventional wing angles were measured at each time step ($t=0.4$ ms) in the body reference frame. $\phi$, wing stroke angle within the stroke-plane; $\eta$, wing deviation angle out of the stroke-plane; $\theta$, wing rotation angle along the spanwise axis; $\vec{u}$, air velocity vector relative to the wing; $\alpha$, angle-of-attack of the wing. (**C**) Wing morphological parameters including their definitions: wingspan $R$, wing surface area $S$, radial position along the span $r$, local wing chord $c$ at distance $r$, the second-moment-of-area $S_2$, and its dimensionless homologue $S_2^*$.

across species, with males exhibiting larger values in some cases and females in others. Interspecific differences in morphology were much stronger (**Supplementary file 1a**), making sexual dimorphism negligible in comparison. While sexual differences in flight kinematics could not be assessed in this study (see Materials and methods), they are likely to be similarly outweighed by interspecies variation.

When testing the phylogenetic signal on wing morphology and body size among the 28 studied species, we detected a significant effect of phylogeny on all traits except for the second-moment-of-area. In contrast, the effect of phylogeny on flight, tested on eight species, was non-significant for all flight traits (**Supplementary file 1b**).

## No correlation between wingbeat kinematics and body mass

We quantified a total of 33 wingbeats across eight hoverfly species, including at least three wing-beats per species (**Figures 2 and 3**). The wingbeat frequencies of the sampled wingbeats were highly correlated with the average frequency across entire flight sequences ($f=186 \pm 43$ Hz; $= 178 \pm 40$ Hz; mean ± standard deviation; $n=33$ flight sequences; $r=0.98$, $p<0.001$). Here, the number of wingbeats per flight sequence was $26 \pm 28$. This suggests that the digitised wingbeats reliably reflect those of the full flight sequences. Furthermore, the mean advance ratio for the digitised wingbeats was below the generally accepted threshold of 0.1 ($J=0.08 \pm 0.02$), confirming that the hoverflies were operating approximately in a hovering flight mode (**Ellington, 1984a**).

The eight hoverfly species exhibited similar wingbeat kinematics (**Figures 2 and 3A–E**). Here, they all moved their wings forwards and backwards sinusoidally within the stroke-plane (**Figure 3A**), with a wing stroke amplitude of $A_\phi=100°\pm18°$ (**Figure 3H**). Movements out of the stroke-plane, as expressed by the deviation angle, are minimal (**Figure 3B**). Therefore, the oscillatory angular speed of the wing-beats (**Figure 3D**) is primarily caused by the wing stroke movement and peaks at approximately $5°\times10^4$ s$^{-1}$ (**Figure 3F**). During the majority of each wingbeat (forward and backward wing stroke), the wing operates at approximately constant wing rotation angle and angle-of-attack (**Figure 3C and**

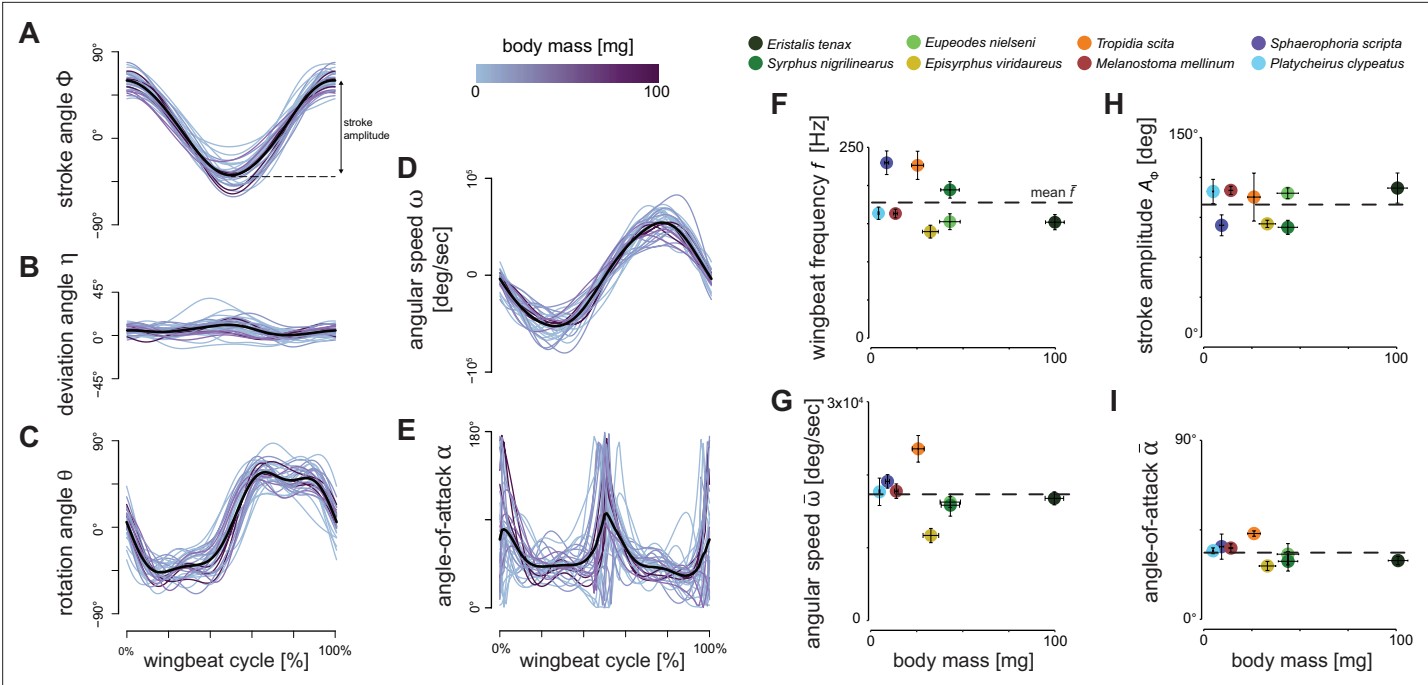

**Figure 3.** Wingbeat kinematics during hovering flight of the eight studied hoverfly species. (**A–E**) Temporal dynamics of the wingbeat kinematics throughout the wingbeat cycle of all digitised wingbeats. Separate wingbeats are colour-coded by body mass (see legend on top), and the black lines show the average wingbeat kinematics for all wingbeats combined. (**A–C**) Temporal dynamics of the three conventional wingbeat kinematics angle (stroke, deviation, and rotation angle, respectively; see **Figure 2** for definitions). (**D, E**) Angular speed and angle-of-attack of the wings throughout the wingbeat cycle, derived from wing stroke, deviation, and rotation angles. (**F–I**) Derived wingbeat-average wingbeat kinematics parameters for each studied species versus the body mass of that species. The kinematics parameters are wing stroke amplitude $A_\phi$, wingbeat frequency $f$, wingbeat-average angular wing speed $\bar{\omega}$, and mean angle-of-attack at mid wing stroke $\bar{\alpha}$, respectively. Each data point shows mean ± standard error per species and is colour-coded according to the legend on the top. None of the wingbeat-average wingbeat kinematic parameters were significantly associated with body mass (**Table 1**). Horizontal dashed lines show the mean parameter values, as expected under kinematic similarity.

The online version of this article includes the following figure supplement(s) for figure 3:

**Figure supplement 1.** Derived wingbeat-average wing kinematics parameters for each studied species versus the body mass of that species.

**E**, respectively). At the end of the forward and backward wing stroke, the wings rapidly supinate and pronate, respectively. This causes the wing to flip upside down at stroke reversal, allowing it to operate again at the constant wing angle-of-attack of $\bar{\alpha}=42°\pm10°$ (**Figure 3I**). The hoverflies flew with a mean body pitch angle of $\beta_{body}=30°\pm16°$. The corresponding pitch angle of the wing stroke-plane was $\beta_{stroke-plane}=-15°\pm16°$, showing that these hoverflies hover with an inclined stroke-plane.

The temporal dynamics of wingbeat kinematics was apparently not associated with variation in body mass (**Figure 3A–E**). Accordingly, none of the derived wingbeat-average wing kinematics parameters varied significantly with body mass, although a small negative trend was observed for the wingbeat frequency (**Table 1**, **Figure 3**, **Figure 3—figure supplement 1**). These include the parameters that directly affect aerodynamic force production (**Equations 1 and 2**), wingbeat-average angular speed $\bar{\omega}$, angle-of-attack $\bar{\alpha}$, stroke amplitude $A_\phi$ and wingbeat frequency $f$, and the additionally estimated wingbeat kinematic parameters (rotation amplitude $A_\theta$, deviation amplitude $A_\eta$, and peak stroke rate $\dot{\phi}_{peak}$, rotation rate $\dot{\theta}_{peak}$, and deviation rate $\dot{\eta}_{peak}$) (**Supplementary file 1d**).

Furthermore, variations in wingbeat kinematics were little affected by the flight kinematics metrics flight speed and climb angle (**Supplementary file 1c**). Only flight speed had a positive significant effect on the angular speed of the beating wings, suggesting that faster flying hoverflies beat their wings at higher angular speeds. When controlling for this effect of flight speed on angular wing speed, the regression between angular wing speed and body mass remained non-significant ($R^2=2\%$; $p=0.41$). Additionally, flight speed was not correlated with body mass ($R^2=1\%$, $p=0.43$).

Altogether, these findings contradict our second null hypothesis, which proposed that hoverflies maintain weight support through allometric scaling of wingbeat kinematics (with wing morphology

**Table 1.** Results of phylogenetic generalised least squares (PGLS) regressions of the log10-transformed morphological and wingbeat kinematic parameters from the aerodynamic model (*Equations 1 and 2*) relative to log10-transformed body mass.

Estimated scaling factors of which the 95% confidence interval exclude the scaling factor for geometric similarity are indicated in bold and with a star, suggesting allometric scaling of that metric (see last column). The preceding two columns show the scaling factor for geometric or kinematic similarity, and the scaling factor for maintaining weight support across sizes via allometric changes of the specific parameter, assuming all other parameters scale under geometric and kinematic similarity.

| | $n$ | $p$ | $R^2$ | Intercept | Scaling factor estimate [95% CI] | Scaling factor for similarity | Scaling factor for single-metric weight support | Allometry |
|---|---|---|---|---|---|---|---|---|
| **Wing morphology** | | | | | | | | |
| Second-moment-of-area $S_2$ | 28 | <0.001 | 86% | −2.949 | **1.008 [0.767 - 1.250]** * | 4/3=1.33 | 1 | Negative |
| Wingspan $R$ | 28 | <0.001 | 84% | −0.479 | **0.255 [0.192 - 0.317]** * | 1/3=0.33 | 2/9 = 0.22 | Negative |
| Wing chord $\bar{c}$ | 28 | <0.001 | 88% | −1.145 | 0.294 [0.229 - 0.359] | 1/3=0.33 | 0 | Isometry |
| Normalised second-moment-of-area $S_2^*$ | 28 | <0.001 | 34% | −0.413 | **−0.034 [−0.051 - −0.018]** * | 0 | −1/3 = −0.33 | Negative |
| **Wingbeat kinematics** | | | | | | | | |
| Wingbeat frequency $f$ | 8 | 0.427 | 9% | 2.343 | −0.084 [−0.161 - −0.063] | 0 | −1/6 = −0.17 | – |
| Stroke amplitude $A_\Phi$ | 8 | 0.785 | 1% | 2.041 | −0.017 [−0.134 - −0.101] | 0 | −1/6 = −0.17 | – |
| Wing angular speed $\bar{\omega}$ | 8 | 0.225 | 0% | 10.161 | −0.101 [−0.249 - −0.046] | 0 | −1/6 = −0.17 | – |
| Angle-of-attack $\bar{\alpha}$ | 8 | 0.105 | 22% | 3.947 | −0.085 [−0.174 - −0.002] | 0 | −1/3 = −0.33 | – |

scaling isometrically), i.e., that the product of wingbeat frequency and amplitude should scale with mass under negative allometry ($\omega \sim f\ A_\phi \sim m^{-1/6}$). Variation in wingbeat kinematics between species thus cannot be explained by the demands of maintaining weight support throughout (isometric) size changes (*Figure 3*, *Figure 3—figure supplement 1*). Allometric adjustments for maintaining weight support across sizes among hoverfly species thus more likely occur through changes in wing morphology, as formulated in our first null hypothesis.

## Wing morphology covaries with body mass

Wing morphology strongly covaried with body mass: the wing's second-moment-of-area ($S_2$), span ($R$), mean chord ($\bar{c}$), and normalised second-moment-of-area ($S_2^*$) all significantly correlated with body mass (*Figure 4*, see also *Figure 4—figure supplement 1*). Covariation between the wing morphological parameters and body mass all deviated from the isometric expectation, except for mean chord (*Table 1*, *Figure 4*). Second-moment-of-area, wingspan, and $S_2^*$ all showed a significant negative allometry, i.e., they were all disproportionately larger in smaller species (*Figure 4A–C*, *Table 1*), supporting our first null hypothesis that weight support is maintained across sizes using allometric scaling of wing morphology.

Scaling of the second-moment-of-area with size differed the most from isometric scaling for morphological similarity ($a_{allo} = 1.01$ versus $a_{sim} = 1.33$) and almost perfectly aligned with the scaling for metric-specific weight support ($a_{ws} = 1$). The corresponding relative scaling factor confirms this (*Equation 3*), as it is close to 100% ($a_{S2}^* = 103\%$; *Figure 4E*). This suggests that hoverfly species of different size should be able to maintain weight support during hovering flight almost purely through variations in the second-moment-of-area.

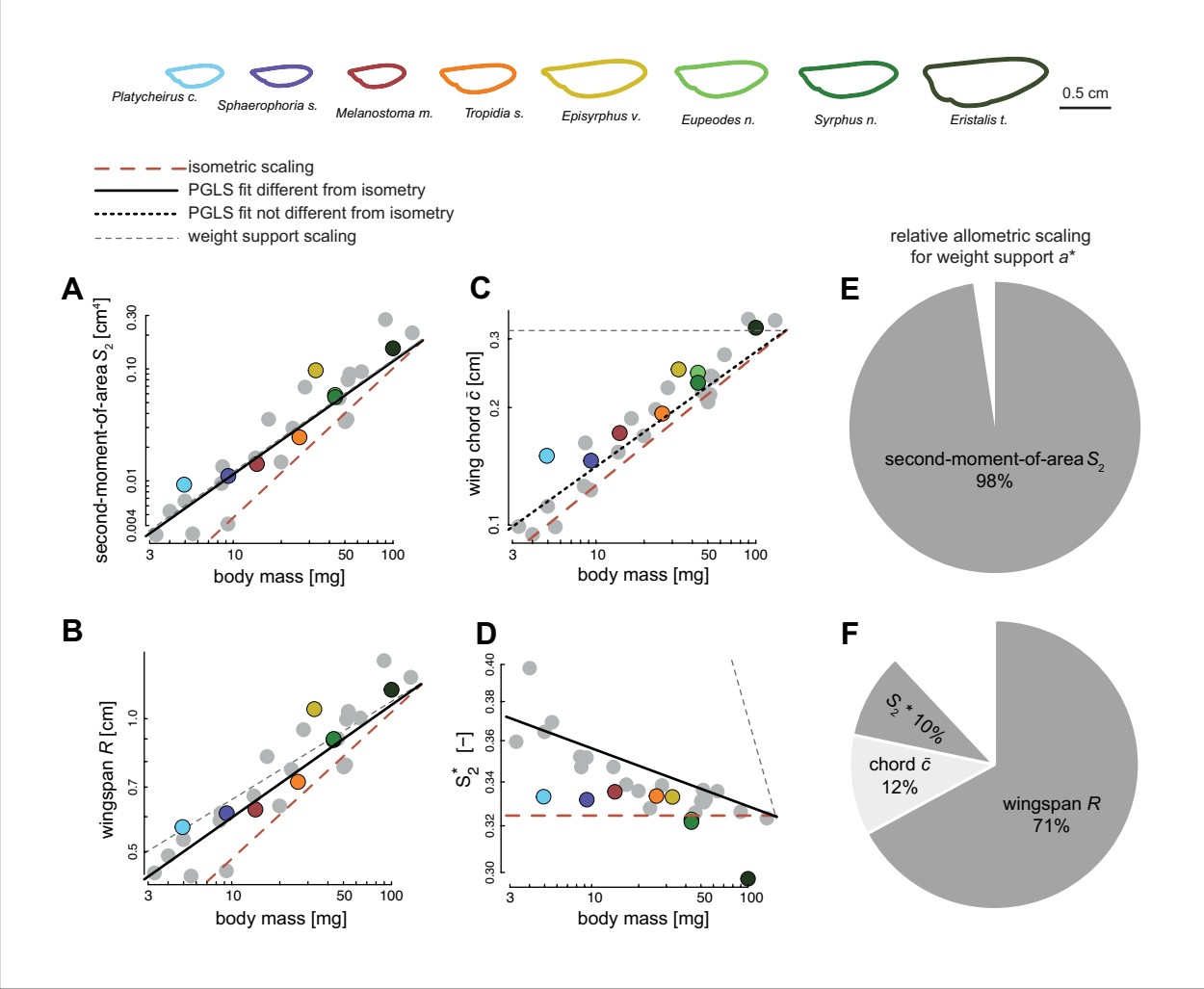

**Figure 4.** The scaling of wing morphology with body mass for all 28 studied hoverfly species (**A–D**), and how allometric scaling of wing morphology contributes to maintaining weight support across the hoverfly size range (**E, F**). (**A–D**) Body mass (abscissa) versus on the ordinate the second-moment-of-area $S_2$, wingspan $R$, mean chord $\bar{c}$, and normalised second-moment-of-area $S_2^*$, respectively. Each data point shows a species average value. Data points of species for which flight was studied are colour-coded (see top row), and species with only quantified morphology are shown in grey (see the corresponding individual-specific data in *Figure 4—figure supplement 1*). The black, red, and grey trendlines show the best fitting line from a phylogenetic generalised least squares (PGLS) regression, isometric scaling, and the expected scaling for metric-specific maintenance of weight support, respectively (see legend above A). All fitted PGLS regression slopes, except for wing chord, are significantly lower than expected under geometric similarity, suggesting negative allometric scaling of wing size and shape with respect to body mass (see also *Table 1*). (**E, F**) The relative contributions of different wing morphology metrics to maintaining weight support across the studied range of hoverfly sizes, expressed by the relative allometric scaling factor $a^*$. (**E**) The relative allometric scaling factor for the second-moment-of-area $S_2$ and (**F**) the relative allometric scaling factor for the separate morphological components of $S_2$: wingspan $R$, mean chord $\bar{c}$, and normalised second-moment-of-area $S_2^*$.

The online version of this article includes the following figure supplement(s) for figure 4:

**Figure supplement 1.** Wing morphology parameters versus body mass for all studied specimens.

By estimating the relative scaling factors of the various morphological parameters that compose the second-moment-of-area ($a_R^* = 71\%$; $a_c^* = 12\%$; $a_{S2^*}^* = 10\%$, *Figure 4F*), we find that adaptations in wingspan contribute the most to the variations in second-moment-of-area. Allometric scaling in the normalized second-moment-of-area was smaller yet significant, indicating a notable contribution of change in wing shape to the adaptations in second-moment-of-area across sizes.

Our phylogenetic geometric morphometrics analysis revealed two main axes of wing shape variation among the studied species associated with the principal components PC1 and PC2, explaining 69% and 15% of the variation, respectively (*Figure 5—figure supplements 1 and 2*). Shape variation

carried on PC1 was positively correlated with the $S_2^*$ (r=−0.49; p=0.009) and negatively with body mass (r=0.55; p=0.002) (*Figure 5A and B*, respectively), depicting the changes in wing shape associated with body mass variation among hoverfly species. This change in $S_2^*$ reflects a wing surface area mostly located distally in smaller species, whereas most wing surface area is located near the wing base in larger hoverfly species (*Figure 5*).

The combined correlative and morphometrics analyses thus show that as their size decreases, hoverflies exhibit relatively larger wings that are also shaped such that their normalised second-moment-of-area is increased (*Figures 4 and 5*). These combined morphological adaptations allow smaller hoverflies to maintain in-hovering weight support without the need to adjust their wingbeat kinematics.

## Aerodynamic modelling confirms the main role of wing morphology for weight support

Because among the eight hoverfly species, wingbeat kinematics did not vary significantly with body mass (*Table 1*), we performed all CFD simulations with the average wingbeat kinematics of all species combined (*Figure 6A*). We did vary the wing shape and size between species (top row of *Figure 6*), and we modelled aerodynamic force production with both the species-specific wingbeat frequency and mean wingbeat frequency of all species combined (*Figure 6B*).

The resulting aerodynamic forces differed between species mostly in magnitude, but the temporal dynamics of force production were strikingly similar (*Figure 6C*, *Figure 6—figure supplement 1*). For all species, the temporal dynamics of the vertical aerodynamic force showed a large force peak preceded by a small peak during each wing stroke (forward and backward stroke). The backstroke produces a larger vertical force and a more distinct higher harmonic. These dynamics are similar between species, despite their different wing shapes. This suggests that wing shape does not strongly affect the characteristics of aerodynamic force production.

The wingbeat-average vertical aerodynamic forces showed a strong scaling with body mass (*Figure 6D*), as expected due to the differences in wing size between species (top row in *Figure 6*). For each studied species, we modelled aerodynamic forces, both at the species-specific wingbeat frequency and at the mean wingbeat frequency of all species (*Figure 6B and D*). For the simulations with species-specific wingbeat frequency, the hoverflies produced on average weight support, whereas for the mean wingbeat frequency cases, weight support was not achieved (*Figure 6D*). Variation in wingbeat frequency thus appeared necessary for individual hoverflies to achieve weight support, although between species wingbeat frequencies do not correlate with body mass.

We continued to analyse the relative effect of variations in wing morphology and wingbeat frequency on the aerodynamic force production, as quantified from CFD (*Figure 7*, *Table 2*). Here, all the tested morphology and kinematics parameters varied significantly with aerodynamic force magnitude, except for wingbeat frequency and normalised second-moment-of-area.

First, we correlated the vertical forces computed using CFD with the product of second-moment-of-area and wingbeat-average angular speed squared (*Figure 7A*). According to our quasi-steady aerodynamic model, this should yield a linear correlation (*Equation 1*: $\bar{F} \sim S_2 \bar{\omega}^2$), which was also the case as the allometric scaling factor of $a_{allo} = 0.99$ [0.97–1.01] (median [95% confidence interval]; n=8) is not significantly different from linear (*Table 2*). The corresponding allometric scaling factor equals 103%, suggesting that the allometric adaptations that drive variations in $S_2 \bar{\omega}^2$ fully explain variations in aerodynamic force production required for maintaining weight support across sizes. It also shows that our quasi-steady aerodynamic model predicts the effect of morphology and kinematics on aerodynamic force production well.

Separating this dependency of aerodynamic forces on $S_2 \bar{\omega}^2$ into its primary components, second-moment-of-area and wingbeat frequency (*Figure 7B–D*) show that allometric adaptations in second-moment-of-area and wingbeat frequency contribute 81% and 22% to maintaining weight support across sizes, respectively. Hereby, the aerodynamic forces do not scale significantly with wingbeat frequency (*Figure 7B*; *Table 2*), as was also the case for the equivalent correlation of wingbeat frequency with body mass (*Figure 3F*; *Table 1*). These combined results confirm that adaptations in morphology drive variations in aerodynamic force production between hoverfly species of various sizes, although variations in wingbeat frequency cannot be ignored.

Further parting the effect of $S_2$ on aerodynamic force production into its components (*Figure 7E–G*) shows that allometric adaptations in wingspan $R$, mean chord $\bar{c}$, and normalised second-moment-of-area

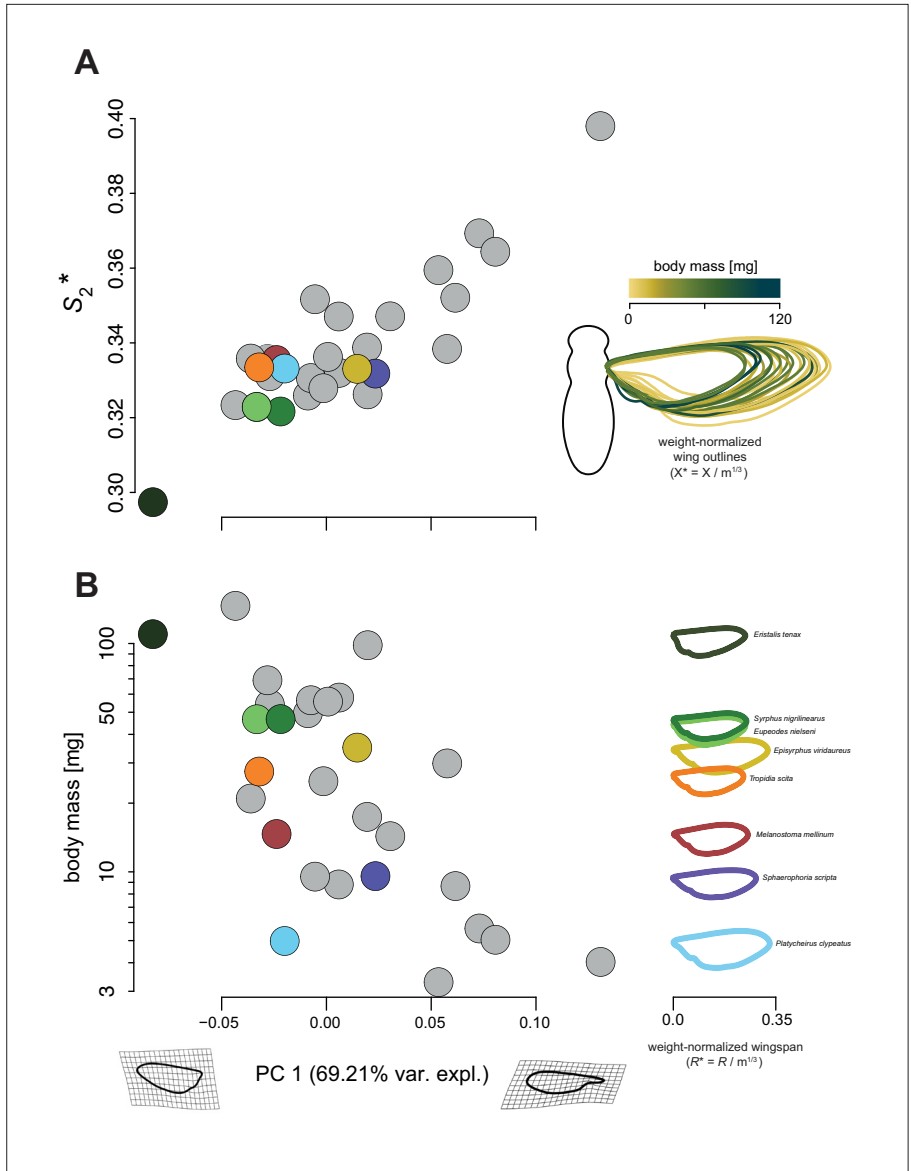

**Figure 5.** Changes in wing shape and size associated with body mass variation. Each data point shows a species-average value (see the individual-specific data in *Figure 5—figure supplement 2*), and is colour-coded for species of which both flight and morphology were studied (see right column of B), and in grey for species with only morphology quantified. (**A**) Normalised second-moment-of-area ($S_2^*$) versus the primary principal component (PC1) of the phylogenetic principal component analysis (phyloPCA) performed on the mean wing shape coordinates of all 28 studied hoverfly species. On the right, we show the 28 wing outlines colour-coded and normalised with body mass (with coordinates $X^* = X/m^{1/3}$). (**B**) Body mass (m) versus PC1 (left), and mass-normalised wing outlines versus body mass for the eight species used in our flight experiments (right). The left and right gritted wing shapes on the PC1 axis show the theoretical wing shapes at maximum and minimum value of PC1. PC1 explains 65.21% of the variations in wing shape (see also *Figure 5—figure supplements 1 and 2*). (**A, B**) The combined results show that in larger species, wing surface area was located more proximally (lower PC1 and $S_2^*$ values) than in smaller species, in which wing area tended to be located more distally (higher PC1 and $S_2^*$ values). Combined with the changes in wing shape (left), weight-normalised wingspan ($R^*$) and mean chord ($\bar{c}$) tend to be larger in smaller species (right).

The online version of this article includes the following figure supplement(s) for figure 5:

**Figure supplement 1.** Result of geometric morphometrics analysis on the wing outlines of 28 hoverfly species; data per species.

**Figure supplement 2.** Result of geometric morphometrics analysis on the wing outlines of 28 hoverfly species; data per individual.

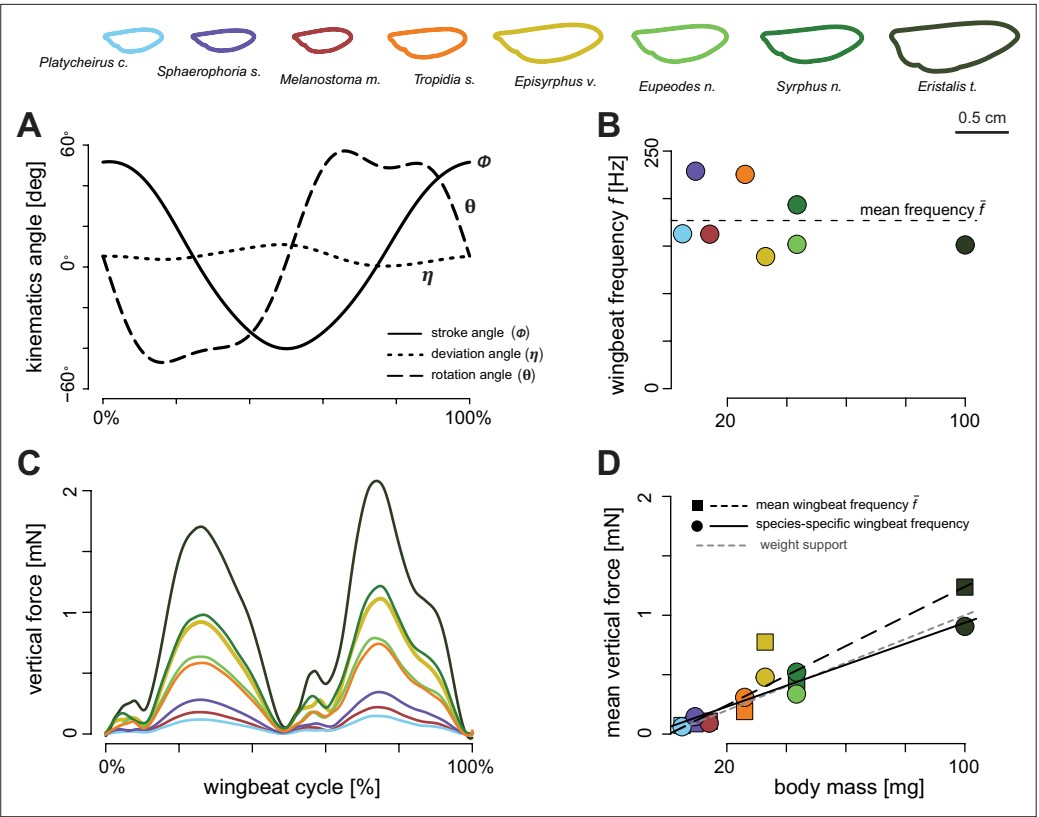

**Figure 6.** Aerodynamic forces produced by hovering hoverflies, as estimated using computational fluid dynamics (CFD) simulations. (**A, B**) For our simulations, we used the species-specific wing shapes and sizes (top row), but average wingbeat kinematics and frequency ($\bar{f}$) across all eight studied hoverfly species (**A, B**). (**C**) The resulting temporal dynamic of vertical forces throughout the wingbeat cycle, coloured by species (see legend on top). (**D**) The wingbeat-average vertical force versus body mass, for all simulated hoverfly species operating at the average wingbeat frequency for all species $\bar{f}$ (square data points and dashed trend line), and forces scaled to the species-specific wingbeat frequency $f$ (round data points in B and D, and solid trend line). The trendline for weight support ($F=mg$) is shown with a grey dashed line. With the species-specific wingbeat frequency $f$ (solid trend line and circles in B and D) hoverflies are closer to producing weight support than for the simulations at the average wingbeat frequency $\bar{f}$ (dashed lines in B and D).

The online version of this article includes the following figure supplement(s) for figure 6:

**Figure supplement 1.** Temporal dynamics of the vertical aerodynamic forces produced during the wingbeat of the eight studied hoverfly species, estimated using computational fluid dynamics (CFD) simulations.

---

$S_2^*$ contribute 55%, 19%, and 9% to variations in aerodynamic force production across sizes, respectively (**Figure 7H**). Here, normalised second-moment-of-area did not significantly scale with vertical force production (**Figure 7G**; **Table 2**), as opposed to the significant scaling found in the corresponding analysis with body mass for the extended 28 species dataset (**Figure 4D**; **Table 1**).

The sum of the relative scaling factors of all metrics contributing to vertical force production (**Equation 2**) equals 105% (**Figure 7H**), suggesting that these allometric adaptations combined fully explain variations in aerodynamic force production required for maintaining weight support across sizes. The here-identified effects of morphological traits on CFD-based forces are strikingly similar to those from the comparative analysis between morphology and body mass (**Figure 4**). This shows that our kinematics analysis based on eight species captures the major effects of wing morphology on aerodynamic force production, as expected based on the morphological analysis with the large sample size of 28 species.

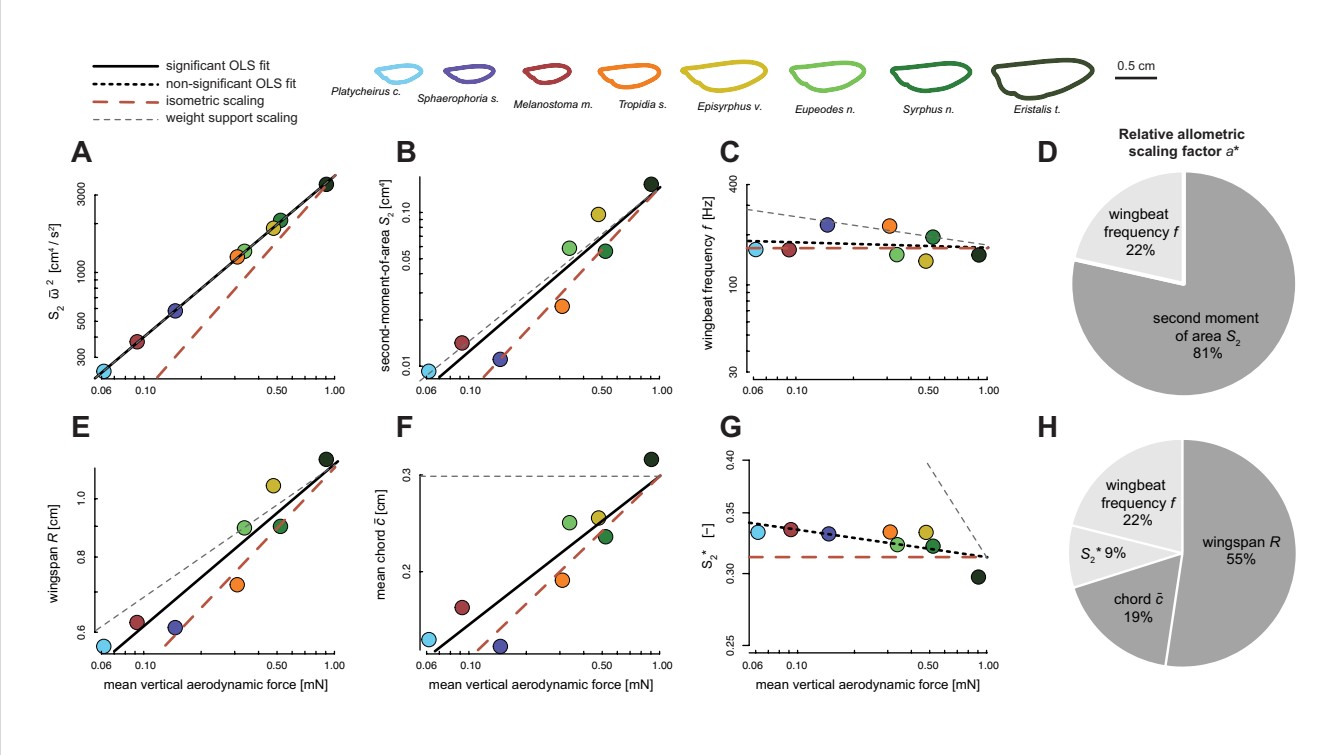

**Figure 7.** The relative contribution of allometric variations in wing morphology and wingbeat kinematics to computational fluid dynamics (CFD)-derived vertical aerodynamic force production, and how this contributes to maintaining in-hovering weight support across the sampled range of hoverfly sizes. (A–C, E–G) Wingbeat-average vertical force at species-specific wingbeat frequency f (abscissa) versus on the ordinate various wing morphology and kinematics traits. In each panel, data points are results for different species, colour-coded according to the legend on the top. The solid and dashed black lines show the significant and non-significant ordinary least squares (OLS) regression fits, respectively. The dashed red and grey lines show the expected slopes for scaling under morphological and kinematic similarity and for 100% metric-specific weight support, respectively. (D, H) The relative contributions of allometric variations in wing morphology and kinematics to maintaining weight support across the studied range of hoverfly sizes, expressed by the relative allometric scaling factor a* (*Equation 3*). Contributions based on significant and non-significant OLS regressions are shown in dark and light grey, respectively. (A) The wingbeat-average vertical force scales linearly with the product of second-moment-of-area and wingbeat-average angular speed squared (*Equation 1*: $F \sim S_2 \, \bar{\omega}^2$), resulting in weight support across all sizes. (B–D) Separating this product into its main components (B and C, respectively) shows that the second-moment-of-area and wingbeat frequency contribute 81% and 22% to maintaining weight support across sizes, respectively (D). (E–H) Parting the contribution of second-moment-of-area into its components (E–G) shows that allometric variations in wingspan, mean chord, and normalised second-moment-of-area contribute 55%, 19%, and 9% to maintaining weight support across sizes, respectively (H).

## Discussion

### Allometric changes in wing morphology enable small hoverflies to maintain weight support during hovering

In this study, we investigated whether variation in body mass among hoverfly species is associated with changes in wing morphology (null hypothesis one) or in wingbeat kinematics (null hypothesis two). Both components directly influence the aerodynamic forces produced by beating wings during flapping flight (*Ellington, 1984c*) and therefore could accommodate for maintaining weight support during evolutionary changes in size occurring throughout the diversification of hoverfly species. Comparing how wingbeat kinematics and wing morphology scale with body mass across hoverfly species, we showed that wing size and shape rather than wingbeat kinematics scale with body mass. This supports the first null hypothesis while rejecting the second, suggesting that wing morphology might respond stronger than wingbeat kinematics to the constraints imposed for maintaining weight support during flight in hoverflies.

Changes in wing morphology associated with body mass variation are commonly observed in flying animals (*Le Roy et al., 2019*; *Norberg, 2007*; *Rayner, 1988*; *Taylor and Thomas, 2014*; *Tercel et al., 2018*), indicating that wing and body morphology evolve together as an integrated phenotype.

**Table 2.** Results of ordinary least squares (OLS) regression between the log10-transformed vertical aerodynamic force estimated from computational fluid dynamics (CFD) and log10-transformed morphological and kinematics parameters, for all eight hoverfly species studied using CFD.

OLS regressions for metrics that scale significantly with aerodynamic force magnitude are indicated with a star, and have p-values in bold (**p<0.05**). For each tested parameter, we estimated its relative contribution to maintaining weight support across sizes using the relative allometric scaling factor $a^*$ (**Equation 3**), based on the estimated scaling factor, the scaling factor for geometric or kinematic similarity, and the expected scaling for maintaining metric-specific weight support.

| | $n$ | p | $R^2$ | Intercept | Scaling factor estimate [95% CI] | Scaling factor for similarity | Scaling factor for metric-specific weight support | Relative allometric scaling factor $a^*$ |
|---|---|---|---|---|---|---|---|---|
| $S_2 \cdot \bar{\omega}^2$ | 8 | **<0.001***  | 100% | 3.593 | 0.989 [0.974 - 1.005] | 4/3 = 1.33 | 1 | 103% |
| Second moment of area $S_2$ | 8 | **<0.001***  | 88% | −0.844 | 1.063 [0.667 - 1.461] | 4/3 = 1.33 | 1 | 81% |
| Wingbeat frequency $f$ | 8 | 0.667 | 3% | 2.218 | −0.037 [−0.238 - 0.164] | 0 | −1/6 = −0.17 | 22% |
| Wingspan $R$ | 8 | **<0.001***  | 88% | 0.058 | 0.272 [0.173 - 0.371] | 1/3 = 0.33 | 2/9 = 0.22 | 55% |
| Wing chord $\bar{c}$ | 8 | **0.001***  | 83% | −0.525 | 0.271 [0.147 - 0.394] | 1/3 = 0.33 | 0 | 19% |
| Normalised second moment of area $S_2^*$ | 8 | 0.055 | 48% | −0.504 | −0.031 [−0.061 - 0.000] | 0 | −1/3 = −0.33 | 9% |

Increasing size is usually not only associated with relatively larger wings, but also accompanied by changes in wing shape (**Danforth, 1989**; **Outomuro et al., 2013**). Teasing apart the changes in wing morphology resulting from constraints imposed by body mass from that of other selective pressures is, however, not straightforward. It is generally facilitated by comparing species with similar ecology, because this limits the effect of contrasted ecological pressures at play in different species (**García and Sarmiento, 2012**). Hoverfly species present a similar ecology in that all adults are specialised flower visitors (**Klecka et al., 2018**). Comparable selective regimes are thus likely acting on the evolution of flight performance and associated flight kinematics and morphology in adult hoverflies.

Shared ancestry can also affect the evolution of wing morphology, and its effect should thus be controlled to understand how body mass may constrain the evolution of other traits (**Blomberg et al., 2003**). Among the 28 species studied here, variation in the measured morphological traits was significantly influenced by phylogenetic history. In contrast, among the sub-set of eight species for which we studied flight kinematics, phylogenetic signal in flight traits was non-significant. This may suggest either a higher evolutionary lability in flight compared to morphological traits, or results from limited statistical power due to the small number of species with quantified flight data. Note that the lack of phylogenetic signal does not necessarily preclude an impact of phylogeny on the covariation between traits and body mass. As such, changes in wing morphology (and possibly in wingbeat kinematics) resulting from constraints imposed by body mass are confounded with the effect of shared ancestry. Our phylogenetically informed regressions of morphological and flight traits against body mass nonetheless allow disentangling such effects.

The large size variation among hoverfly species results from 90 million years of evolutionary diversification (**Wong et al., 2023**). While some taxa may have evolved larger body sizes than their ancestors, others may have undergone size reduction. Determining the exact trajectory of body size evolution in the species studied is beyond the scope of our study. Addressing the aerodynamic challenges of size reduction may not fully represent the evolutionary constraints acting on all hoverfly species. However, we chose to focus on the adaptation of the wing-based propulsion system in this context, as it provides a valuable framework for understanding the evolution of flight in small flying insects.

As size decreases, the wings become relatively less effective in producing the aerodynamic forces required for weight support (see Appendix 1). The constraints imposed by weight support during hovering flight should then promote the evolution of disproportionately large wings, or an increased wingbeat frequency or amplitude in smaller hoverfly species. We found that none of the wingbeat kinematic parameters significantly scaled with body mass, but that wings indeed tended to be disproportionately larger in smaller species (i.e. negative allometry). Here, size-dependent adaptations in the primarily wing morphology parameter second-moment-of-areas ($S_2$; **Equation 1**) almost fully explained the required adaptations for maintaining weight support across sizes ($a_{S2}^* = 98\%$; **Figure 4E**). Parsing

the second-moment-of-areas in its separate wing size and shape components shows that both the primary wing size and wing shape metrics (wingspan $R$ and normalised second-moment-of-areas $S_2^*$, respectively) exhibit significant negative allometric scaling (*Figure 4B and D*). These allometric adaptations in wing size and shape are predicted to contribute to maintaining weight support across sizes for 71% and 10%, respectively (*Figure 4F*). Thus, changes in multiple aspects of the wing morphology may contribute to mitigating the negative effect of size reduction on flight ability, whereby wing shape effects are small but significant (*Figure 5*).

A negative allometry between wing size and body size has been found in other insect groups: for solitary bees, this was found at both the interspecific and intraspecific level (*Grula et al., 2021*), and at the intraspecific level in rhinoceros beetles (*Kawano, 1995*). In contrast, disproportionately larger wings were found in larger species of dragonfly, suggesting positive allometry in these larger insects (*May, 1981*). This opposite allometric scaling of wing size in dragonflies and hoverflies may be the result of the use of contrasted mechanisms for producing aerodynamic forces, involving both morphological or kinematic variations, and contrasted flight modes (e.g. gliding flight in dragonflies).

The observed negative allometry in wing shape with mass was further emphasised in our geometric morphometrics analysis (*Figure 5*, *Figure 5—figure supplement 1*, and *Figure 5—figure supplement 2*). Indeed, in our phylogenetic principal component analysis (phyloPCA) performed on superimposed wing shape coordinates, the primary axis PC1 explained a striking 69% of the variations in wing shape and was strongly correlated with normalised second-moment-of-area $S_2^*$. This PC1 shape variation depicts a shift in the spanwise distribution of wing area, from more proximally located in larger species to more distally located in smaller species (*Figure 5*). Interestingly, similar changes in wing shape associated with decreasing body size have been found among species of Hymenoptera (*Danforth, 1989*). An illustration of the most pronounced end of this pattern is the highly spatulated wing shape found in tiny wasp species (e.g. in the Mymaridae or Mymarommatidae families). This shows that although the relative contribution of allometric scaling of wing shape to maintaining weight support is relatively small (10%; *Figure 4F*), physical constraints associated with size reduction are strong enough to drive these adaptations. Evolutionary changes in the body size of hoverflies may thus have favoured wing shapes in smaller species that produce aerodynamic forces more effectively.

We continued to study how allometric variations in wing morphology and wingbeat kinematics affected aerodynamic force production across the studied range of hoverfly sizes (*Figures 6 and 7*). For this, we combined flight experiments on eight hoverfly species with both analytical quasi-steady aerodynamic modelling (*Equations 1 and 2*) and numerical aerodynamics modelling (CFD). By combining these results, we first tested the validity of our quasi-steady aerodynamics model by correlating the wingbeat-average vertical forces estimated from CFD with the product of second-moment-of-area and wingbeat-average angular wing speed ($F \sim S_2\,\bar{\omega}^2$; *Equation 1*; *Figure 7A*). The resulting linear correlation shows that our quasi-steady aerodynamic model effectively captures the effect of wing morphology and kinematics on aerodynamic force production. By separating the combined effect of variations in morphology and kinematics on aerodynamic force production into its components (*Equation 2*), we confirm that adaptations in morphology drive the variations in aerodynamic force production between hoverfly species of different sizes. Here, wing size variations as expressed by wingspan have the largest effect on aerodynamic force production, but variations in wing shape and wingbeat frequency cannot be completely ignored (*Figure 7C, G and H*). In fact, the respective 9% and 22% contribution of wing shape and wingbeat frequency to variations in aerodynamic force production needs to be included in our model to maintain 100% weight support across the size range of hovering hoverflies (*Figure 7H*).

## Factors decoupling wingbeat kinematics from the influence of body mass

Our study revealed that wingbeat kinematics is relatively stable across hoverfly species, with none of the measured kinematics parameters significantly correlated with body mass (*Figure 3*; *Table 1*). Performing CFD simulations using the average wingbeat kinematics for all species, but incorporating differences in wing morphology, demonstrated that weight support was indeed achieved across species. However, it is important that variation in wingbeat frequency – albeit uncorrelated with body mass – has proven crucial for correctly estimating the aerodynamic force necessary to achieve weight support (*Figures 6D, 7C and D*). Adjustments in wingbeat frequency thus appear to play a role in

achieving weight support during hovering for individual hoverflies. The hovering hoverflies might thus primarily use wingbeat frequency to finely tune the vertical aerodynamic force required for weight support. Unlike birds and bats, flying insects cannot actively morph their wings to control aerodynamic force production (*Harvey et al., 2022*). As a result, flying insects rely purely on wingbeat kinematics changes to trim aerodynamic forces and torques during steady flight (*Muijres et al., 2017*) and to adjust these forces and torques during manoeuvring flight (*Hedrick et al., 2009*; *Muijres et al., 2014*). Our CFD simulation results suggest that hovering hoverflies use relatively simple wingbeat frequency modulations to compensate for the continuous daily fluctuations in body mass, as a result of food and water intake and excretion.

The lack of covariation between in-flight wing movement and body mass goes against the prediction that wingbeat kinematics is more likely to be adjusted with varying body mass due to the greater flexibility of flight kinematics as compared to morphology. This finding is also surprising given that size variation is generally accompanied by changes in wingbeat kinematics in other flying animals (*Norberg, 2007*; *Rayner, 1988*; *Riskin et al., 2010*; *Taylor and Thomas, 2014*; *Tercel et al., 2018*). A previous study dissecting the flapping wing kinematics of miniaturised insects found an increased wing deviation angle in smaller species, enabling higher wing velocity and hence aerodynamic force production (*Lyu et al., 2019*). Such a change in wing kinematics associated with decreasing size can be explained by the increased viscous drag constraint that tiny flying insects experience at low Reynolds numbers, requiring higher aerodynamic force generation. Here, we did not find an increased wing deviation angle in smaller hoverfly species, most likely because our study focuses on a too narrow range of sizes and consequently too similar Reynolds numbers. Our CFD analysis confirms this, as the effect of Reynolds number aerodynamic force production was negligible.

While only a handful of studies have examined the effect of body mass on detailed wingbeat kinematics in insects (such as wingbeat amplitudes and angle-of-attack, *Belyaev et al., 2014*; *Lyu et al., 2019*; *Meresman and Ribak, 2017*), the scaling between body mass and wingbeat frequency has been frequently assessed (*Bartholomew and Casey, 1978*; *Byrne et al., 1988*; *Casey et al., 1985*; *Dudley, 1990*; *Tercel et al., 2018*; *de Nadai et al., 2021*). In theory, an increase in wingbeat frequency is expected in smaller insects because flapping wing-based aerodynamic force production decreases faster than body mass. As such, it is puzzling that the reduction in size among the studied hoverfly species is not associated with increased wingbeat frequency.

Interestingly, inconsistent results on the scaling between body mass and wingbeat frequency have been found among other groups of flying animals. Although predominantly negative (e.g. in butterflies [*Dudley, 1990*], in orchid bees [*Darveau et al., 2005*], in mosquitoes [*de Nadai et al., 2021*], and in a comparative study across insect orders [*Tercel et al., 2018*]), several studies found weak or no relationship between body mass and wingbeat frequency (e.g. in moths [*Bartholomew and Casey, 1978*], whiteflies [*Byrne et al., 1988*], bees [*Duell et al., 2022*; *Grula et al., 2021*]). Consistent with our findings, a study comparing the wingbeat kinematics during tethered flight among hoverfly species spanning a range of body mass similar to that investigated here found no significant effect of body mass on wingbeat kinematics (*Belyaev et al., 2014*).

Interestingly, multiple studies have highlighted that deviation from the expected scaling of wingbeat frequency with body mass is often associated with allometry in wing area (see literature review, *Darveau, 2024*). Flying animals exhibiting negative allometry in wing area tend to have lower wingbeat frequencies than would be predicted based on their body mass alone (*Bartholomew and Casey, 1978*; *Dorsett, 1962*; *Norberg, 2012*). In line with this pattern, the lack of increased wingbeat frequency in small hoverflies is here associated with disproportionately large wings.

Beyond morphological constraints, ecological factors may also contribute to explaining the heterogeneous scaling trends found among insect groups. The extent to which body mass constrains the evolution of wingbeat kinematics may depend on species ecology. Adaptation to a particular ecological niche may drive the evolution of specialised flight abilities, promoting a specific wingbeat kinematics. Selective pressures exerted on these wingbeat kinematics may sometimes overcome the constraints imposed by weight support. In that case, relatively constant wingbeat kinematics would be maintained over a range of body mass, and changes in wing morphology alone are expected to adjust for weight support. Consistent with this hypothesis is the decoupling of wingbeat kinematics from body mass variation, which has been highlighted in species where hovering flight ability is crucial for food acquisition. These include bees (*Duell et al., 2022*; *Grula et al., 2021*), sphingid moth (*Bartholomew*

and Casey, 1978), hummingbirds (Skandalis et al., 2017), and hoverflies (Belyaev et al., 2014) as also in the present study. The absence of correlation between body mass and wingbeat kinematics observed in hoverflies may thus be due to their highly specialised hovering flight abilities.

The selective pressures promoting the evolution of unique hovering flight abilities in hoverflies are not precisely known but hovering steadily is likely advantageous for navigating between flowers when foraging for nectar. Good hovering flight performance may also have been promoted because it increases mating opportunities during the lek gathering of males (Downes, 1969; Gilbert, 1984; Heinrich and Pantle, 1975). Thus, selection for increased foraging efficacy and/or mating opportunities may have maintained consistent wingbeat kinematics across hoverfly species despite variation in body mass.

To further ascertain our findings, quantifying sexual dimorphism in hoverfly wingbeat kinematics would be a relevant future research direction. This would clarify the role of mating behaviour on the evolution of hovering flight performance. In this study, sexes could not be distinguished during our flight experiments, preventing us from testing whether male hoverflies exhibit better hovering performance. Furthermore, additional wingbeat kinematics measurements in species from different habitats would help verify the generality of our results in assessing the possible effect of habitat. Contrasted environmental conditions (e.g. temperature, humidity, or altitude) can affect flight performance (Altshuler and Dudley, 2003; Farnworth, 1972) and thus interfere with the scaling between flight and morphology.

Here, we focused on how variations in wing morphology and kinematics allow hoverflies of different sizes to produce weight support in hovering flight. Hovering flight is a functionally and energetically demanding type of locomotion (Ellington, 1991), but other flight behaviours might be even more so (Alexander and Vogel, 2004; Muijres et al., 2014). Future research should also focus on how the variation in scaling highlighted here influences such peak locomotor performance, especially because the effect of morphology on performance is generally most pronounced when individuals are pushed to their limits (Losos et al., 2002). For instance, investigating how the differential scaling of wing morphology and kinematics with size affects peak thrust production during evasive flight manoeuvres would be a relevant experiment for future study.

Finally, a critical but overlooked aspect of our study is the interplay between wingbeat kinematics and muscle physiology. Although our analysis focuses on the wing-based propulsion system, wing motion is ultimately powered by the muscular motor. The scaling of muscle mechanical output is linked to wingbeat kinematics, yet the findings discussed above are largely decoupled from the muscle constraints that may accompany changes in body size. Future studies that integrate muscle physiology with aerodynamic modelling are thus needed to complement the current approach.

## Conclusion

Altogether, our results suggest that the hovering flight abilities of hoverflies may stem from highly specialised wingbeat kinematics that have been largely conserved throughout their diversification. In contrast, changes in wing morphology have evolved with the aerodynamic constraints associated with evolutionary changes in size. The existence of selective pressures maintaining stable wingbeat kinematics despite variation in body mass implies that high hovering flight performance can only be achieved over a limited range of wingbeat kinematic parameters, which remains to be ascertained.

Our study, moreover, highlights that, unlike many other Diptera, hoverflies employ a highly stereotyped, inclined stroke-plane wingbeat pattern for hovering, where during the forward dorsal-ventral wing stroke the wing moves not only from back to front but also downwards (Mou et al., 2011). This wingbeat kinematics pattern may contribute to the specialised hovering abilities of hoverflies, as it is consistently observed among the eight species studied here, with their stroke-plane pitch angle during hovering being approximately orientated at −30°.

Furthermore, despite significant changes in wing morphology and wingbeat frequency among the studied species, the temporal dynamics of aerodynamic force production is strikingly similar between them (Figure 6C, Figure 6—figure supplement 1). This suggests that not only the wingbeat kinematics is conserved between species, but also the underlying aerodynamics including the relative use of different unsteady aerodynamic mechanisms (Sane, 2003). Thus, the apparent highly specialised

flight style of hoverflies is maintained among several species with a large range of body masses and robust against variations in wing shape and wingbeat frequency.

## Materials and methods

**Key resources table**

| Reagent type (species) or resource | Designation | Source or reference | Identifiers | Additional information |
|---|---|---|---|---|
| Biological sample (*Ceriana conopsoides*) | 4 individuals | Leiden Biodiversity Center | | |
| Biological sample (*Dorylomorpha* sp.) | 3 individuals | Leiden Biodiversity Center | | |
| Biological sample (*Episyrphus viridaureus*) | 3 individuals | Wild (Wageningen University surroundings) | | |
| Biological sample (*Eristalinus aeneus*) | 4 individuals | Leiden Biodiversity Center | | |
| Biological sample (*Eristalis tenax*) | 10 individuals | Wild (Wageningen University surroundings) | | |
| Biological sample (*Eupeodes nielseni*) | 4 individuals | Wild (Wageningen University surroundings) | | |
| Biological sample (*Helophilus pendulus*) | 4 individuals | Leiden Biodiversity Center | | |
| Biological sample (*Leucozona lucorum*) | 4 individuals | Leiden Biodiversity Center | | |
| Biological sample (*Leucozona nigripila*) | 4 individuals | Leiden Biodiversity Center | | |
| Biological sample (*Melangyna guttata*) | 4 individuals | Leiden Biodiversity Center | | |
| Biological sample (*Melangyna lasiophthalma*) | 4 individuals | Leiden Biodiversity Center | | |
| Biological sample (*Melanostoma dubium*) | 3 individuals | Leiden Biodiversity Center | | |
| Biological sample (*Melanostoma mellinum*) | 10 individuals | Wild (Wageningen University surroundings) | | |
| Biological sample (*Microdon analis*) | 3 individuals | Leiden Biodiversity Center | | |
| Biological sample (*Myolepta dubia*) | 4 individuals | Leiden Biodiversity Center | | |
| Biological sample (*Neoascia annexa*) | 4 individuals | Leiden Biodiversity Center | | |
| Biological sample (*Neoascia geniculate*) | 4 individuals | Leiden Biodiversity Center | | |
| Biological sample (*Pipizella viduata*) | 4 individuals | Leiden Biodiversity Center | | |
| Biological sample (*Pipunculus campestris*) | 3 individuals | Leiden Biodiversity Center | | |
| Biological sample (*Platycheirus albimanus*) | 4 individuals | Leiden Biodiversity Center | | |
| Biological sample (*Platycheirus clypeatus*) | 3 individuals | Wild (Wageningen University surroundings) | | |
| Biological sample (*Psilota atra*) | 4 individuals | Leiden Biodiversity Center | | |
| Biological sample (*Senaspis elliotti*) | 3 individuals | Leiden Biodiversity Center | | |
| Biological sample (*Sphaerophoria scripta*) | 4 individuals | Wild (Wageningen University surroundings) | | |
| Biological sample (*Syrphus nigrilinearus*) | 6 individuals | Wild (Wageningen University surroundings) | | |
| Biological sample (*Tropidia scita*) | 4 individuals | Wild (Wageningen University surroundings) | | |
| Biological sample (*Volucella hyalinipennis*) | 3 individuals | Leiden Biodiversity Center | | |
| Biological sample (*Volucella inanis*) | 4 individuals | Leiden Biodiversity Center | | |
| Software, algorithm | R | R | RRID:SCR_001905 | R version 4.1.2 |
| Software, algorithm | MATLAB | MATLAB | RRID:SCR_001622 | MATLAB version 2021b |

## Collecting and identifying hoverflies

We quantified body and wing morphology in 28 hoverfly species, of which we collected eight species in the wild and twenty from museum specimens at the Naturalis Biodiversity Center (Leiden, the Netherlands). The wild-caught live hoverflies were also used for flight experiments, to quantify their wingbeat kinematics during hovering flight (*Figure 1*).

From the museum collection, we sampled 74 individual specimens, aiming to include two males and two females per species whenever possible. This resulted in 4.2 ± 1.7 individuals per species (mean ± standard deviation). Museum specimens were selected to maximise both phylogenetic coverage and size range. The eight hoverflies from the wild were collected in September 2022 on the campus of Wageningen University, the Netherlands (51°59′01.0″N 5°39′32.7″E; ca. 10 m). We captured a total of 44 individuals, resulting in a sample size of 5.5 ± 2.9 individuals per species. Flies were captured with hand-held nets and brought in the lab to record their flight within an hour after capture, after which morphology was quantified.

The wild species were first distinguished visually, but their species identity was subsequently ascertained using DNA barcoding. This was done by sending a piece of leg of each individual to the Canadian Centre for DNA Barcoding (CCDB), where barcoding was performed following standard automated protocols of the BOLD Identification System (http://www.boldsystems.org/) (*Ratnasingham and Hebert, 2007*). Sequenced DNA was then compared with a reference library to establish a species match. Sexes were determined visually using criteria from *Gilbert, 2015*.

## Flight experiments

The flight experiments were carried out in a custom-built octagonal flight arena made of transparent Plexiglas (50×50×48 cm$^3$, height ×width×length; *Figure 2A*; *Cribellier et al., 2022*). Stereoscopic high-speed videos were recorded using three synchronised high-speed cameras (Photron FASTCAM SA-X2), equipped with Nikon Sigma 135 mm lenses, and recording at a temporal resolution of 5000 frames s$^{-1}$, a spatial resolution of 1084×1084 pixels, and a shutter speed of 1/10,000 s. The cameras were positioned around the arena to provide a top view and inclined left- and right-side view of the flying insects, which resulted in a zone of focus of approximately 12 cm$^3$ in the centre of the arena in which body and wing movement were in view and in focus of all three cameras. To increase contrast and therefore facilitate subsequent tracking, the cameras were back-lit using three infrared light panels, placed behind the bottom and lower side walls of the octagon arena, opposite each camera (*Figure 2A*).

Prior to filming, the stereoscopic camera system was calibrated with the direct linear transformation (DLT) technique (*Hartley and Sturm, 1995*), by digitising a wand of 2.8 cm long moved throughout the zone of focus of the cameras. To track the wand movement, we trained an artificial neural network using the pose estimation Python library DeepLabCut (*Nath et al., 2019*), achieving a test error of 3.1 pixels. Computation of the DLT coefficients was then done using the MATLAB program easyWand (*Theriault et al., 2014*). The calibration root mean square error was 1.21 pixels.

During the flight experiment, all the individuals from a single species were released together in the arena to increase the probability that an individual flew through the intersecting fields of view of the three cameras. Most individuals tended to remain on the sidewalls, occasionally making short flights across the arena. For each species, the filming experiment was carried out until a minimum of five suitable flight sequences were obtained, whenever possible. A suitable sequence was defined as a flying individual clearly crossing the intersecting fields of view. Although this procedure facilitated the recording, this prevented identifying which specific individual or sex was recorded flying, leaving open the possibility of recording the same individual multiple times. Note that this approach aimed at estimating the average wingbeat kinematics per species as we could not directly assign flight measurements at the individual level.

## Wingbeats selection

First, we determined the flight speed throughout each recorded flight sequence. We did this by tracking the hoverfly body centre, estimated from the mean position between the head and abdomen extremities, throughout the flight sequence (mean duration of the tracked trajectories = 0.13 ± 0.11 s) using a trained DeepLabCut neural network (*Nath et al., 2019*) (test error: 6.3 pixels). From this, we calculated flight speed using a central temporal differentiation scheme. Based on these speed data,

we selected flight sequences with the lowest average speed and, within each sequence, identified the wingbeat occurring during the slowest part of the trajectory. For this selection process, we only included the flight trajectory sections in which the hoverfly was flying straight at a low climb angle, to prevent possible variation in wingbeat kinematics stemming from flight manoeuvres. We digitised a minimum of three wingbeats per species, each taken from a different flight sequence. For six of the species we tracked, three wingbeats, and for two species (*Melanostoma mellinum* and *Sphaerophoria scripta*), we tracked six wingbeats. Note that our experiment did not allow capturing perfect hovering flight sequences (i.e. flight speed = 0 m s$^{-1}$) but aimed at getting as close as possible to this flight mode. This was evaluated using the advance ratio (see next section).

## Quantifying wingbeat kinematics

We quantified the wing movement using the manual stereoscopic video tracker *Kine* in MATLAB (*Fontaine et al., 2009*; *Fry et al., 2003*) originally designed to track the wingbeat kinematics of fruit flies. The tracker was tailored to hoverflies by building wing models matching the wings of the studied hoverfly species. Wing models were obtained by digitising the wing outline on high-resolution photographs and consisted of 41 2D coordinates representing the wing shape, with the known hinge and tip positions. To reduce complexity, wings were considered as rigid flat plates, although real hoverfly wings endure deformations during the wing stroke (*Walker et al., 2010*).

We tracked the wingbeat kinematics on downsampled videos (from 5000 to 2500 frames s$^{-1}$), as this preserved high enough temporal resolution to capture the wing movement ($n$=27 ± 5 frames per wingbeat). The tracking provided us with the position and orientation of the body and wings in each consecutive video frame. The position and orientation of the body were defined in the world reference frame, as a 3D position vector ($\mathbf{X}$=[$x,y,z$]) and the consecutive Euler angles body yaw, pitch, and roll. The angular orientation of the wings was expressed in the hoverfly body reference frame (*Muijres et al., 2014*; *Figure 2B*), as Euler angles relative to the stroke-plane: the stroke angle ($\phi$), deviation angle ($\eta$), and rotation angle ($\theta$) of each wing (*Figure 2B*). The stroke-plane of each wing was defined as the plane at a 45° angle to the long axis of the body passing through the hinge position of that wing. The three wing kinematics angles (stroke, deviation, and rotation angle) were measured separately on the left and right wing and then averaged between the two wings. The time history of the averaged angles was then fitted with a fourth-order Fourier series for all three angles (*Muijres et al., 2014*), and their temporal derivatives were computed (i.e. stroke, deviation, and rotation rate). We estimated the absolute angular speed of the beating wing as $\omega = \sqrt{\dot{\phi}^2 + \dot{\eta}^2}$, where $\dot{\phi}$ and $\dot{\eta}$ are the stroke rate and deviation rate, respectively. The angle-of-attack ($\alpha$) was computed as the angle between the wing plane and the velocity vector of the wing (*Figure 2B*).

Based on the temporal dynamic of the wing kinematics angles, we derived the following wingbeat-average kinematics parameters: wingbeat frequency, defined as $f$=1/$T_{\text{wingbeat}}$, where $T_{\text{wingbeat}}$ is the duration of the wingbeat; wing stroke, deviation, and rotation amplitudes, defined as $A_\phi$=|$\phi_{\max}-\phi_{\min}$|, $A_\eta$=|$\eta_{\max}-\eta_{\min}$|, and $A_\theta$=|$\theta_{\max}-\theta_{\min}$|, respectively; we estimated the wingbeat-average angular speed as $\bar{\omega} = 2fA_\phi$; we estimated the mean angle-of-attack $\bar{\alpha}$ as the mean value at mid forward and backward wing stroke, where aerodynamic force production is close to maximum (*Tian et al., 2013*).

We quantified the wingbeat-average body kinematics during each wingbeat using the mean flight speed, climb angle, and the pitch angle of both body and stroke-plane. Flight speed $\mathbf{U}$ was estimated as the temporal derivative of body positions, and climb angle as $\gamma_{\text{climb}}$=atan($U_{\text{ver}}/U_{\text{hor}}$), where $U_{\text{ver}}$ and $U_{\text{hor}}$ are the vertical and horizontal component of the flight velocity, respectively. Body pitch angle $\beta_{\text{body}}$ was calculated as the angle between the body axis and the horizontal. The equivalent stroke-plane pitch angle was defined as the pitch angle of the stroke-plane relative to the horizontal, calculated as $\beta_{\text{stroke-plane}}$=$\beta_{\text{body}}$–45°.

To verify whether the studied wingbeats were representative of the overall recorded flight behaviour, we calculated the mean wingbeat frequency over the full flight sequence as $\hat{f} = n_{\text{wingbeats}}/T_{\text{sequence}}$, where and $T_{\text{sequence}}$ are the number of wingbeats in a sequence and the sequence duration, respectively. This mean wingbeat frequency was then compared with that of the studied wingbeats. To estimate whether the analysed wingbeats were representative for hovering flight, we calculated the advance ratio of all wingbeats as $J = U/(\bar{\omega}R)$, where $R$ is the span of a single wing. Hovering is generally defined as flight with advance ratios $J$<0.1, i.e., when the forward flight speed is less than 10% of the wingbeat-induced speed of the wingtip (*Ellington, 1984a*).

Finally, although hoverfly flight speed was minimised in the wingbeat selection procedure, remaining variation in body kinematics may still affect the wingbeat kinematics and therefore confound with changes in wingbeat kinematics resulting from scaling effect. We therefore assessed if the flight kinematics metrics flight speed and climb angle affected wingbeat kinematics by performing multiple regression analyses. Here, each wingbeat kinematic metric was treated as a dependent variable and flight speed and climb angle as model effects.

## Quantifying morphology

We quantified body and wing morphology in both the wild-caught hoverflies and the museum specimens. After being filmed flying, wild-caught individuals were sacrificed by freezing them at –20°C and weighed with a resolution of 0.1 mg using an analytical balance (XSR204 Mettler Toledo). Wings were then detached from the body and photographed using a digital microscope (Zeiss Stemi SV11). From the museum collection, we only selected specimens with properly flattened (i.e. non-folded) wings and photographed them using a Nikon D600 camera equipped with a 60 mm lens. Here, we photographed the complete animal from above, and the separate wing by placing parallel to the camera lens. We estimated body mass of museum specimens using the correlation between thorax width and the fresh mass in the wild-caught specimens (*Figure 1—figure supplement 1*).

From the wing photographs of both the wild and museum specimens, we measured the primary wing morphology parameters using WingImageProcessor in MATLAB (available at https://biomech.web.unc.edu/wing-image-analysis/). These include single-wing area $S$, wingspan $R$, mean wing chord $\bar{c}$, and second-moment-of-area $S_2$. From these, we calculated weight-normalised wingspan as $R^* = R/m^{1/3}$, and the normalised second-moment-of-area as $S_2^* = S_2/(R^3 \bar{c})$. The weight-normalised wingspan is a parameter describing relative wingspan, and the normalised second-moment-of-area is defining wing shape changes independent of changes in wing size. Both parameters allowed us to identify variations in relative wingspan and relative wing shape between species of different sizes, respectively.

Next, we used a landmark-based geometric morphometric method to quantify more precisely the variation in wing shape between species (*Bookstein, 1997*). Wing shape outline was determined using 300 semi-landmarks equidistantly spaced along the wing outline. Semi-landmarks are commonly used to describe curvy shapes such as wing outlines, which lack typical identifiable landmarks (*Gunz and Mitteroecker, 2013*). Semi-landmarks are allowed to slide along the curve to establish geometric correspondence (*Gunz and Mitteroecker, 2013*). After this alignment, the landmarks are treated as regular ones in subsequent analyses. We placed one fixed landmark at the base of the wing, fixing the overall landmark configuration with respect to this homologous position available for all specimens. All landmarks were digitised using *TpsDig2* (*Rohlf, 2015*). Wing outlines were superimposed by performing a generalised Procrustes analysis (*Rohlf and Slice, 1990*) using the *geomorph* R package (*Adams and Otárola-Castillo, 2013*). This procedure isolates the shape information from the extraneous variations, namely the size, position, and orientation. To visualise wing shape variation, we then performed a phyloPCA on the superimposed coordinates using the *phytools* R package (*Revell, 2012*). Shape changes carried on the PCA axes were visualised using the *plotRefToTarget* function in geomorph.

To assess the strength of morphological sexual dimorphism and whether it may interfere with interspecific differences, we tested the effect of species and sex on the morphological parameter using analyses of variance (ANOVAs).

## Testing the influence of phylogeny on flight kinematics and morphological traits

To assess whether shared evolutionary history among the studied species influenced variation in morphology and wingbeat kinematics, we tested if closely related species showed more similar values in the measured parameters than distantly related ones. This was done by computing phylogenetic signals with the Blomberg's K-statistics (*Blomberg et al., 2003*) on each morphological and flight parameter using the R package *phytools* (*Revell, 2012*). Furthermore, we accounted for the effect of phylogeny when assessing the scaling relationships between size and traits using phylogenetic generalised least squares (PGLS) regression (*Symonds and Blomberg, 2014*). For the PGLS regressions, we assumed a Brownian motion model of evolution, and the model parameters were estimated using maximum likelihood. Tests were performed on the mean values per species using the most recent phylogeny of hoverflies (*Wong*

*et al., 2023*). The phylogeny was pruned to align with our sample of 28 species for the morphological analysis and further pruned to include only the eight species used in the flight analysis.

## Modelling the aerodynamics of hoverfly flight

To estimate how changes in wing morphology and wingbeat kinematics affect the aerodynamic forces required for flight, we modelled the aerodynamic vertical force produced by flapping wings in hovering flight using a simplified quasi-steady approach as (*Ellington, 1984c*)

$$\bar{F} = \frac{1}{2} \rho \, S_2 \, \bar{\omega}^2 \, \bar{C}_F \,, \tag{1}$$

where $\bar{F}$ is the wingbeat-average upward-directed aerodynamic lift force produced by a beating wing. This aerodynamic force thus varies quadratically with the angular speed of the beating wing ($\bar{\omega}^2$), and linearly with air density ($\rho$), the second-moment-of-area of the wing ($S_2$), and the lift force coefficient ($\bar{C}_F$).

The second-moment-of-area $S_2$ is the only morphological parameter in this equation, thus capturing how wing morphology affects aerodynamic force production, from a first principle. But this second-moment-of-area parameter combines both wing size and shape characteristics (i.e. wing area and its distribution over the spanwise axis; *Ellington, 1984b*). To disentangle the respective effect of wing shape and size on aerodynamic force production, the second-moment-of-area can be decomposed as $S_2 = S_2^* R^3 \, \bar{c}$ , where $R$ is the wingspan, $c$ is the mean wing chord, and $S_2^*$ is the normalised second-moment-of-area of the wing. Here, wingspan and average wing chord are size-dependent parameters; the normalised second-moment-of-area describes wing shape as the distribution of area along the spanwise wing axis, independent of wing size.

The lift force coefficient depends on the angle-of-attack, and for fruit flies, this dependency can be modelled as $C_F = \sin(\alpha) \, C_{F\alpha}$, where $C_{F\alpha}$ is the angle-of-attack-specific lift force coefficient (*Dickinson and Muijres, 2016*). For hoverflies, we here assume a similar scaling of lift with angle-of-attack. Both angular speed ($\omega$) and the angle-of-attack ($\alpha$) of the beating wing are wingbeat kinematics parameters that directly affect aerodynamic force production. A flying insect can vary the angular speed of its beating wings by changing both its wingbeat frequency $f$ and wing stroke amplitude $A_\phi$, as angular speed scales linearly with the product of these ($\omega \sim f A_\phi$). The decomposed model for estimating the effect of wing morphology and kinematics of aerodynamic lift force production in flapping insect flight is thus

$$\bar{F} = \frac{1}{2} \, \rho \, R^3 \, \bar{c} \, S_2^* \left( f A_\phi \right)^2 \sin \left( \bar{\alpha} \right) \bar{C}_{F\alpha}. \tag{2}$$

We used this model to decompose the relative effect of variations in wingbeat kinematic and wing morphology between species on aerodynamic force production.

## Physical scaling of kinematics and morphology parameters with size

In hovering flight, the aerodynamic vertical force $F$ should equal the weight of the animal. Thus, to maintain weight support among different sizes of hoverflies, the aerodynamic force should scale linearly with body mass. This cannot be achieved through isometric scaling, i.e., the expected scaling if geometric similarity is preserved.

For isometric scaling of morphology with mass, all length parameters scale with mass as $l_{iso} \sim m^{1/3}$, resulting in an isometric scale factor under morphological similarity $a_{sim} = 1/3$. As a result, the morphological parameters of interest scale with body mass as $R \sim m^{1/3}$, $\bar{c} \sim m^{1/3}$, and $S_2^* \sim m^0$, and thus both $S_2$ and $F$ scale with body mass as $F \sim S_2 \sim m^{4/3}$ (see Appendix 1 for detailed derivation). This value is larger than one, and so with an isometric reduction in size, the aerodynamic force for weight support reduces more quickly than body mass. Thus, compared to large species of hoverflies, small species need relatively larger wings or adjust their wingbeat kinematics to maintain weight support.

For the wingbeat kinematics, we define kinematic similarity when the size-independent kinematics parameters of interest also do not scale with size and should thus remain constant across body mass. Specifically, wingbeat frequency ($f$), angular stroke amplitude ($A_\phi$), angular speed ($\omega$), and angle-of-attack ($\alpha$) are all expected to scale under kinematic similarity as $f \sim m^0$, $A_\phi \sim m^0$, $\omega \sim m^0$, and $\alpha \sim m^0$, respectively. As hoverflies become smaller, deviations from this kinematic similarity might occur to

adjust for the reduction in weight support due to isometric size reduction. We emphasise that kinematic similarity is a pure hypothetical scenario used as a baseline prediction for comparing variations in flight kinematics between species and sizes of hoverflies. This approach facilitates the identification of deviations from scale-invariant wingbeat kinematics required for producing weight support during hovering flight (see the 'Theoretical consideration' section for further details).

Using our aerodynamic force model (*Equations 1 and 2*), we can estimate how the combination of wing morphology and wingbeat kinematics should scale with size, while maintaining weight support for hovering flight across sizes ($\bar{F} = mg$). Here, we do so for two extreme cases: (1) weight support is maintained across sizes using allometric scaling of wing morphology only, and thus wingbeat kinematics are kept constant (kinematic similarity); (2) weight support is maintained across sizes using allometric scaling of wingbeat kinematics, while wing morphology scales isometrically. These two scenarios are thus two alternative null hypotheses to be tested in this study:

1. To maintain weight support with non-varying wingbeat kinematics, the second-moment-of-area should scale linearly with body mass ($S_2 \sim m$), resulting in negative allometry of $S_2$ relative to body mass ($S_2 \sim m^1 < m^{4/3}$). This could be achieved by negative allometric scaling in the wingspan, mean chord, or normalised second-moment-of-area ($R < m^{1/3}$, $\bar{c} < m^{1/3}$, or $S_2^* < m^0$), or a combination of these. To maintain weight support via negative allometric scaling of only one of these three metrics, while the others scale isometrically, then these should scale with mass as $R \sim m^{2/9}$, $\bar{c} \sim m^0$, or $S_2^* \sim m^{-1/3}$ (see Appendix 1 for detailed derivations).

2. To maintain weight support via allometric scaling of kinematics, while wing morphology scales isometrically ($S_2 \sim m^{4/3}$), the angular speed of the beating wing should scale with body mass as $\omega \sim m^{-1/6}$ (see Appendix 1 for detailed derivations). Again, a hovering hoverfly could achieve this using a comparative adjustment in wingbeat frequency or wing stroke amplitude ($f \sim m^{-1/6}$ or $A_\phi \sim m^{-1/6}$, respectively), or a combination of both ($\omega \sim f A_\phi \sim m^{-1/6}$). Furthermore, the animal could adjust the aerodynamic force vector using an equivalent adjustment in the wing angle-of-attack ($\sin(\alpha) \sim m^{-1/6}$).

To test these two alternative null hypotheses, we examined how wing morphology and wingbeat kinematics scale with body mass in the studied hovering hoverflies, and whether these scalings were isometric or allometric. We did so using PGLS regressions of the morphology and wingbeat-average kinematics metrics in *Equations 1 and 2* ($S_2^*$, $\bar{c}$, $R$, $\bar{\omega}$, $f$, $A_\phi$, and $\bar{\alpha}$) against the body mass ($m$) (*Symonds and Blomberg, 2014*). Measurements were all $\log_{10}$-transformed prior to the regression analysis. For each morphology and kinematics metric, we tested if the observed scaling factor significantly deviates from geometric similarity or kinematic similarity, respectively ($R \sim m^{1/3}$; $\bar{c} \sim m^{1/3}$; $S_2^* \sim m^0$ and $\bar{\omega} \sim m^0$; $f \sim m^0$; $A_\phi \sim m^0$; $\bar{\alpha} \sim m^0$, respectively), by verifying if its 95% confidence intervals excluded the similarity slope.

To better visualise variations in wing shape and size relative to body mass among species, we scaled all wings with the length scale for geometric similarity ($l_{iso} \sim m^{1/3}$) and calculated the corresponding weight-normalised wingspan as $R^* = R/m^{1/3}$. We performed additional PGLS regressions to test if other kinematic adjustments were associated with changes in body mass, in addition to the ones considered in the aerodynamics model (*Equations 1 and 2*). This includes regressions of body mass versus the amplitudes of the additional measured wingbeat kinematics angles ($A_\eta$, $A_\theta$) and the corresponding peak angular rates ($\dot{\phi}_{peak}$, $\dot{\eta}_{peak}$, $\dot{\theta}_{peak}$).

## The relative contribution of allometric scaling in wing morphology and kinematics to maintaining weight support across sizes

We quantitatively tested our two alternative null hypotheses by assessing how the potential allometric changes in morphology and kinematics contribute to maintaining weight support across the studied range of hoverfly sizes. We did so by comparing, for each relevant morphological and kinematics parameter, its observed mass-specific allometric scaling factor ($a_{allo}$) with both the scaling factor for morphologic or kinematic similarity ($a_{sim}$), and the scaling factor for achieving weight support fully by allometric scaling of only that parameter ($a_{ws}$). Hereby, we calculated the relative allometric scaling factor for weight support as

$$a^* = \frac{a_{allo} - a_{sim}}{a_{ws} - a_{sim}} \cdot 100\% \tag{3}$$

This parameter defines the percentage of which the observed allometric scaling of the specific morphological or kinematics parameter contributes to maintaining weight support during hovering flight, across the studied size range of hoverflies (see Appendix 1 for detailed derivations).

## CFD simulations

We performed CFD simulations to quantify aerodynamic force produced by the different flying hoverfly species. These simulations allowed us to explicitly quantify the effect of size, wingbeat kinematics, and morphology on aerodynamic force production. Furthermore, we used the CFD results to test the validity of our simplified quasi-steady aerodynamic model (*Equations 1 and 2*).

Each simulation was performed with a single rigid, flat wing with species-specific contours, and the averaged wingbeat kinematics across all species. The computational domain was $8R{\times}8R{\times}8R$, and we simulated three consecutive wingbeat cycles. The wing has a thickness of $0.0417R$. We assumed hovering flight mode, so no mean inflow was imposed ($U_\infty = 0$ m s$^{-1}$).

We performed the simulations using our in-house open-source code WABBIT (Wavelet Adaptive Block-Based solver for Insects in Turbulence) (*Engels et al., 2022*; *Thomas Engels et al., 2021*). The computational approach is based on explicit finite differences that solve the incompressible Navier-Stokes equation, in an artificial compressibility formulation. Wavelets are used to dynamically adapt the discretisation grid to the actual flow at every given time $t$. This allows us to focus the computational effort where it is required to ensure a given precision, while the grid is significantly coarsened wherever possible. Starting from a coarse base grid, we allow up to seven refinement levels, each one refining the grid by a factor of two. In this work, we use the Cohen-Daubechies-Feauveau wavelet CDF42. The grid is composed of blocks of identical size with $B_s = 22$ points in all three spatial directions, yielding an effective maximum resolution of 352 points along the wingspan. An additional simulation with eight refinement levels confirmed the validity of our results.

As demonstrated in numerous validation tests, obtained numerical data can be regarded as quasi-exact in the sense that they are equivalent to an experiment with identical geometry and kinematics. More details on the code can be found in *Thomas Engels et al., 2021*; the required parameter files to reproduce the simulations presented here can be found in the supplementary information. To accelerate the computation, each simulation used up to 600 CPU cores with 2.7 TB of memory in total.

From a fluid dynamics point of view, changing wing length $R$ or wingbeat frequency $f$ changes the Reynolds number Re (*Batchelor, 1967*). For each species, we performed a simulation using two different values for Re, one at the average Reynolds number for all species, and a second at the species-specific Reynolds number. This was initially done to separate the effect of Reynolds number and wing shape between species. The non-dimensional aerodynamic force was, however, very similar in both cases; hence, the influence of Reynolds number is negligible. In general, the range of Reynolds numbers for the studied hoverfly species is rather narrow (Re $\in$ [324,1310]). We finally assessed the contribution of interspecific variation in wingbeat frequency to force production by rescaling all aerodynamic forces with the square of the species-specific and average wingbeat frequency ratio ($f/\bar{f}$)$^2$.

## The relative contribution of variations in wing size, shape, and kinematics to aerodynamic force production

By combining the CFD simulation results with our aerodynamic force model (*Equations 1 and 2*) and our physical scaling law analysis, we assessed the relative contribution of changes in wing size, shape, and kinematics between the studied species on the aerodynamic force production required for weight support.

Because wingbeat kinematics were found to vary little across species and to be uncorrelated with body mass, we based this analysis on the average wingbeat kinematics of all species combined (see CFD methods above for details). We did assess the contribution of interspecific variation in wingbeat frequency on aerodynamic force production, as this varied most between species. We did so by rescaling all aerodynamic forces from CFD with the square of the species-specific and average wingbeat frequency ratio ($f/\bar{f}$)$^2$. According to the quasi-steady model, the vertical aerodynamic force from CFD should scale with the product of the wing's second-moment-of-area and the wingbeat-mean angular wing speed squared (*Equation 1*: $\bar{F} \sim S_2\,\bar{\omega}^2$). The angular wing speed scales linearly with wingbeat frequency ($\omega \sim f$), and the second-moment-of-wing-area can be decomposed into the wing size and shape parameters wing span, mean chord, and normalised second-moment-of-area as $S_2 = R^3\,\bar{c}\,S_2^*$ (*Equation 2*).

In our CFD-based scaling analysis, we first tested the validity of our aerodynamic model (*Equations 1 and 2*) by correlating the CFD-based vertical aerodynamic force with the product of second-moment-of-area and wingbeat-average angular speed squared (*Equation 1*: $\bar{F} \sim S_2\,\bar{\omega}^2$). For this, we

used an ordinary least squares (OLS) regression on the log-transformed data to quantify the scaling factor between these metrics. We used the regression statistics to test whether the CFD-based force scaled significantly with $S_2\,\bar{\omega}^2$ (with cut-off at $p<0.05$). Based on these results, we then quantified the corresponding relative allometric scaling factor $a*$ (*Equation 3*), to assess how allometric scaling of $S_2\,\bar{\omega}^2$ contributes to changes in aerodynamic force production at different sizes.

Secondly, we estimated the relative contribution of the primary variables in $S_2\,\bar{\omega}^2$ to aerodynamic force production, being second-moment-of-area and wingbeat frequency. We did so by using the same approach as described above, by first determining the scaling of CFD-based aerodynamic force with second-moment-of-area and wingbeat frequency, and secondly by estimating the corresponding relative scaling factors $a*$. Finally, we used this approach to estimate the relative contributions to aerodynamic force production of the $S_2$ components: wingspan, mean chord length, and normalised second-moment-of-area.

This combined analysis allowed us to estimate the relative contribution of the various morphology and kinematics metrics to variations in vertical force production for weight support across sizes and test for adaptation in wing morphology and kinematics that maintain weight support across sizes. Moreover, comparing the calculated weight support percentages in our model with 100% for full weight support allowed us to test the validity of our analysis approach and aerodynamic force model.

## Theoretical considerations

Our approach uses the concepts of morphological and kinematic similarity to define our working null hypotheses for two extreme scenarios in scaling of the flight propulsion system with size. Here, morphological similarity is defined as the case where all morphological parameters scale isometrically with size; kinematic similarity dictates that all wingbeat kinematic parameters that are not size related also do not scale with size (i.e. $f\sim m^0$, $A_\phi\sim m^0$, $\omega\sim m^0$, and $\alpha\sim m^0$). Based on these concepts, we defined two alternative null hypotheses: (1) weight support is maintained across sizes using allometric scaling of wing morphology only, and thus wingbeat kinematics are kept constant (kinematic similarity); (2) weight support is maintained across sizes using allometric scaling of wingbeat kinematics, while wing morphology scales isometrically (morphological similarity).

It is important to note that both these scenarios of morphological and kinematic similarity are pure theoretical concepts that are physically, biomechanically, nor physiologically sustainable across sizes (*Borelli, 1680*; *Hill, 1950*). We here merely use these theoretical scenarios to define and test our null hypotheses. We do so by comparing the variations in morphology and kinematics for the differently sized hoverfly species with those based on our null hypotheses and based on morphological and kinematic similarity.

## Acknowledgements

The authors thank Ilam Bharathi for helpful discussions about insect flight aerodynamics, and Remco Pieters for his help in building and operating the experimental setup. We are grateful to the Naturalis Centre of Leiden for granting access to their Syrphidae collection, and in particular, to Pasquale Ciliberti for his help with handling collection specimens and taking photographs. We thank the three reviewers who greatly helped improve the first versions of the manuscript. This work was supported by an NWO Vidi research grant to FTM (I/VI.Vidi.193.054). The study was also provided with computer and storage resources to TE by GENCI at IDRIS, thanks to the grant 2023-A0142A14152 on the supercomputer Jean-Zay's CSL partition.

## Additional information

### Funding

| Funder | Grant reference number | Author |
| --- | --- | --- |
| Dutch Research Council | I/VI.Vidi.193.054 | Florian T Muijres |

| Funder | Grant reference number | Author |
| --- | --- | --- |
| Grand Équipement National de Calcul Intensif (France) | 2023-A0142A14152 | Thomas Engels |

The funders had no role in study design, data collection and interpretation, or the decision to submit the work for publication.

## Author contributions

Camille Le Roy, Conceptualization, Resources, Data curation, Formal analysis, Investigation, Visualization, Methodology, Writing – original draft, Writing – review and editing; Nina Tervelde, Resources, Data curation, Writing – review and editing; Thomas Engels, Data curation, Formal analysis, Writing – review and editing; Florian T Muijres, Conceptualization, Formal analysis, Supervision, Funding acquisition, Methodology, Project administration, Writing – review and editing

## Author ORCIDs

Camille Le Roy ⓘ https://orcid.org/0000-0003-0421-296X
Florian T Muijres ⓘ https://orcid.org/0000-0002-5668-0653

Reviewer #2 (Public review): https://doi.org/10.7554/eLife.97839.4.sa1
Author response https://doi.org/10.7554/eLife.97839.4.sa2

# Additional files

## Supplementary files

Source data 1. Morphological data at the individual level.

Source data 2. Temporal dynamic of individual wingbeat kinematics and corresponding flight sequence metadata; the average kinematics per species.

Source data 3. Input data for the CFD simulations, including the average wingbeat kinematics, mean wing length and wing shape of the species studied with CFD.

Source data 4. Phylogenetic trees used in the study.

Source data 5. Mean wing shape for all 28 studied species.

Supplementary file 1. Supplementary tables 1a–d. (a) Results of analyses of variance (ANOVAs) testing the effect of sex and species on wing morphology parameters. (b) Phylogenetic signal computed on morphological and flight traits. Species number in the morphology and flight dataset is 28 and 8, respectively. (c) Results from multiple regressions testing correlations between the wingbeat kinematics parameters and body kinematics, expressed by flight speed and climb angle. (d) Results of phylogenetic generalised least squares (PGLS) regressions of additional wingbeat kinematic parameters against body mass.

MDAR checklist

## Data availability

All data generated or analysed during this study are included in the manuscript and supporting files. Code necessary to reproduce the analyses can be freely used from GitHub (copy archived at *Le Roy, 2025*).

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

## Appendix 1

### Aerodynamic model for flapping insect flight

We modelled aerodynamic lift force production by flapping wings (*F*) during hovering flight using a simplified quasi-steady approach as (*Dickinson and Muijres, 2016*).

$$F = \frac{1}{2}\rho\, S_2\, \omega^2\, C_F. \tag{S1}$$

This shows that the aerodynamic force thus varies quadratically with the angular speed of the beating wing ($\omega^2$), and linearly with air density ($\rho$), the second-moment-of-area of the wing ($S_2$), and the lift force coefficient ($C_F$). The second moment-of-area can be decomposed as $S_2 = S_2^*\, R^3\, \bar{c}$, where $R$ is the wingspan, $\bar{c}$ is the mean wing chord, and $S_2^*$ is the normalised second-moment-of-area of the wing (relative to the wing hinge). The angular speed of a beating insect wing scales with the product of wingbeat frequency and wingstroke amplitude ($\omega \sim f A_\phi$). The lift force coefficient depends on the angle-of-attack, which can be modelled as $C_F = \sin(a)\, C_{Fa}$, where $C_{Fa}$ is the angle-of-attack-specific lift force coefficient (*Dickinson and Muijres, 2016*). The decomposed model for estimating the effect of wing morphology and kinematics on the wingbeat-average aerodynamic lift force produced during hovering flight is thus

$$F = \frac{1}{2}\rho\, R^3\, \bar{c}\, S_2^* \left(f A_\phi\right)^2 \sin\left(\alpha\right) C_{F\alpha}. \tag{S2}$$

### Isometric scaling of morphological parameters with body mass

The **wingspan** *R* and the **mean wing chord** $\bar{c}$ are length scales. Under isometric scaling, mass scales with length to the third power. Thus, wingspan and mean chord length scale as $R \propto m^{1/3}$ and $\bar{c} \propto m^{1/3}$, respectively.

The **wing area** *S* is equal to the product of wingspan and mean chord ($S = R\,\bar{c}$). Substituting the scaling relationships for wingspan and mean chord into this equation gives

$$S \propto m^{1/3} \cdot m^{1/3}$$

$$S \propto m^{2/3}$$

The **second-moment-of-area** $S_2$ scales with the wingspan and mean chord as $S_2 = \bar{c}\, R^3$. Substituting the scaling relationships wingspan and mean chord into this equation gives

$$S_2 \propto m^{1/3} \cdot \left(m^{1/3}\right)^3$$

$$S_2 \propto m^{4/3}$$

The **non-dimensional second-moment-of-area** $S_2^*$ is a wing shape parameter, which is independent of size ($S_2^* = S_2/\bar{c}R^3$). Thus, under isometry, it does not scale with mass as

$$S_2^* \propto m^{4/3}/m^{4/3}$$

$$S_2^* \propto m^0.$$

### Scaling of wingbeat kinematic parameters with body mass under kinematic similarity

Under kinematic similarity, the wingbeat kinematic parameters **angular speed ω**, **wingbeat frequency** *f*, **wingstroke amplitude** *A*, and **angle-of-attack** $\alpha$ do not scale with size and thus should also remain constant across body mass as

$$\omega \propto m^0; f \propto m^0; A_\phi \propto m^0; \alpha \propto m^0.$$

## Expected allometric scaling of morphology and kinematics with body mass for weight support

For weight support during flight, the upward-directed aerodynamic force should balance the weight of the animal ($F = mg$), thus the aerodynamic force should scale with body mass $F \propto m$. Furthermore, our aerodynamic model states that this force scales with wing morphology and kinematics as

$$F \propto \rho R^3 \, \bar{c} \, S_2^* \, \omega^2 \sin(\alpha) \propto \rho R^3 \, \bar{c} \, S_2^* \, (fA_\phi)^2 \sin(\alpha).$$

Based on these scaling laws, we can estimate how specific wing morphology and wingbeat kinematics parameters should scale with mass, given that the other parameters scale isometrically.

### Expected allometric scaling of morphology with body mass for maintaining weight support

The aerodynamic force scales with wing morphology parameters as $F \propto S_2 \propto S_2^* \bar{c} R^3$, and for weight support $F \propto m$. In isometry, these wing morphology parameters scale with mass as $R^3 \propto m$, $\bar{c} \propto m^{1/3}$, and $S_2^* \propto m^0$. Based on this, we find the following allometric scaling of wing morphology for weight support:

The expected scaling of the **second-moment-of-area** $S_2$ for weight support, given that the kinematics parameters scale isometrically, is $S_2 \propto F \propto m$. Thus, to maintain weight support via allometric scaling of the wing's second-moment-of-area only, this should scale linearly with body mass ($S_2 \propto m$).

The expected scaling of **wingspan** $R$ for weight support, when all other parameters scale isometric, can be obtained by isolating $R$ in the above equation, and substituting each element with its scaling relationships

$$m \propto S_2^* \bar{c} R^3$$

$$m \propto m^0 \cdot m^{1/3} \cdot R^3$$

$$R^3 \propto m^{1-1/3} = m^{2/3}$$

$$R \propto m^{2/9}.$$

Thus, to maintain weight support across sizes via allometric scaling of wingspan only, wingspan should scale with body mass as $R \propto m^{2/9}$.

The equivalent expected scaling of the **mean wing chord** $\bar{c}$ for weight support is estimated similarly as

$$m \propto S_2^* \bar{c} R^3$$

$$m \propto m^0 \cdot \bar{c} \cdot \left(m^{1/3}\right)^3$$

$$\bar{c} \propto m^{1-1}$$

$$\bar{c} \propto m^0.$$

Thus, to maintain weight support via allometric scaling of the mean wing chord only, chord length should not scale with mass ($\bar{c} \propto m^0$).

The equivalent expected scaling of the **non-dimensional moment of area** $S_2^*$ for weight support is obtained by

$$m \propto S_2^* \, \bar{c} \, R^3$$

$$m \propto S_2^* \cdot \left(m^{1/3}\right) \cdot \left(m^{1/3}\right)^3$$

$$S_2^* \propto m^{1-4/3}$$

$$S_2^* \propto m^{-1/3}.$$

Thus, to maintain weight support via allometric scaling of the non-dimensional second-moment-of-area only, it should scale with mass as $S_2^* \propto m^{-1/3}$.

### Expected allometric scaling of wingbeat kinematics for maintaining weight support

The aerodynamic force scales with wing morphology and wingbeat kinematics parameters as $F \propto S_2 \omega^2 \alpha \propto S_2 \left(fA_\phi\right)^2 \alpha$, and for weight support $F \propto m$. In isometry, the wing morphology parameter $S_2$ scales with mass as $S_2 \propto m^{4/3}$, and the kinematics parameters do not scale with body mass ($\omega \propto m^0$; $f \propto m^0$; $A_\phi \propto m^0$; $\alpha \propto m^0$). Based on this, we find the following allometric wing kinematics scaling for weight support:

The expected scaling of the **wing angular speed** $\omega$ with body mass for weight support, when all other parameters scale isometric, can be obtained by isolating $\omega$ in the above equation and substituting each element with its specific mass scaling relationship

$$F \propto S_2 \omega^2 \alpha$$

$$m \propto m^{4/3} \cdot \omega^2 \cdot m^0$$

$$\omega^2 \propto m^{-1/3}$$

$$\omega \propto m^{-1/6}.$$

Thus, to maintain weight support across body masses via allometric scaling of the angular wing speed only, it should scale with body mass as $\omega \propto m^{-1/6}$.

The expected scaling of the sub-components of wing speed **wingbeat frequency** $f$ and **amplitude** $A_\phi$ for weight support, while keeping the other component constant, are the same as that of wing speed itself

$$\omega \propto fA_\phi$$

$$A_\phi \propto m^0 \rightarrow f \propto m^{-1/6}$$

$$f \propto m^0 \rightarrow A_\phi \propto m^{-1/6}.$$

Thus, to maintain weight support across body masses via allometric scaling of wingbeat frequency or amplitude only, either of them should scale with body mass as $f \propto m^{-1/6}$ or $A_\phi \propto m^{-1/6}$.

The expected scaling of **angle-of-attack** $\alpha$ for weight support, if everything else scales isometrically with size, can be obtained by isolating $\alpha$ in the equation and substituting each element with its scaling relationships

$$F \propto S_2 \omega^2 \alpha$$

$$m \propto m^{4/3} \cdot m^0 \cdot \alpha$$

$$\alpha = m^{-1/3}.$$

Thus, to maintain weight support via allometric scaling of the angle-of-attack only, it should scale with body mass as $\alpha = m^{-1/3}$.

## The relative contribution of allometric scaling in wing morphology and kinematics to maintaining weight support across sizes

We assessed the relative contribution of allometric scaling of the various wing morphology and wingbeat kinematics parameters to maintaining weight support across sizes. We did so by comparing, for each relevant morphological and kinematics parameter, its observed mass-specific scaling factor ($a_{allo}$) with both the scaling factor for isometry or kinematic similarity ($a_{sim}$), and the scaling factor for achieving weight support fully by allometric scaling of only that parameter ($a_{ws}$). To quantify this contribution for morphological or kinematics parameters $i$, we calculated its relative scaling factor as

$$a_i^* = \frac{a_{allo,i} - a_{,i}}{a_{ws,i} - a_{,i}} \cdot 100\%. \tag{S3}$$

This metric thus estimates the percentage of which the observed allometric scaling of parameter $i$ contributes to maintaining weight support during hovering flight, across the studied size range of hoverflies. The sum of scaling factors of all morphological and kinematics parameters in the aerodynamic force model (*Equation S1 and Equation S2*) combined should equal 100%, to achieve full weight support across the studied range of hoverfly sizes.

$$a_i^* = \frac{a_{allo,i} - a_{,i}}{a_{ws,i} - a_{,i}} \cdot 100\%$$