## [Editor Report · eLife Assessment]

This **important** study addresses how wing morphology and kinematics change across hoverflies of different body sizes. The authors provide **convincing** evidence that there is no significant correlation between body size and wing kinematics across 28 species and instead argue that non-trivial changes in wing size and shape evolved to support flight across the size range. Overall, this paper illustrates the power and beauty of an integrative approach to animal biomechanics and will be of broad interest to biologists, physicists and engineers.

---

## [Referee Report · Reviewer #2 (Public review)]

Summary

Le Roy et al quantify wing morphology and wing kinematics across twenty eight and eight hoverfly species, respectively; the aim is to identify how weight support during hovering is ensured across body sizes. Wing shape and relative wing size vary non-trivially with body mass, but wing kinematics are reported to be size-invariant. On the basis of these results, it is concluded that weight support is achieved solely through size-specific variations in wing morphology, and that these changes enabled hoverflies to decrease in size. Adjusting wing morphology may be preferable compared to the alternative strategy of altering wing kinematics, because kinematics may be subject to stronger evolutionary and ecological constraints, dictated by the highly specialised flight and ecology of the hoverflies.

Strengths

The study deploys a vast array of challenging techniques, including flight experiments, morphometrics, phylogenetic analyses, and numerical simulations; it so illustrates both the power and beauty of an integrative approach to animal biomechanics. The question is well motivated, the methods appropriately designed, and the discussion elegantly places the results in broad biomechanical, ecological, and evolutionary context. In many ways, this work provides a blueprint for work in evolutionary biomechanics; the breadth of both the methods and the discussion reflects outstanding scholarship.

Weaknesses

The work presents a mechanical analysis that is focused solely on aerodynamics; but these aerodynamic demands impose no less relevant demands on the primary engine that drives wing movement: muscle. The relation between the assumed null hypotheses, the observed empirical allometric relations, and the power and work demand they place on muscle remains unclear. Though this is clearly a minor weakness, future work will have to address the link between aerodynamics, wing shape, wing dynamics, and musculoskeletal system in more detail, as discussed briefly by the authors.

---

## [Author Response]

The following is the authors’ response to the previous reviews.

**Reviewer #1 (Public review):**
The paper is well written and the figures well laid out. The methods are easy to follow, and the rational and logic for each experiment easy to follow. The introduction sets the scene well, and the discussion is appropriate. The summary sentences throughout the text help the reader.The authors have done a lot of work addressing my previous concerns and those of the other Reviewers.

We are pleased that the revised manuscript satisfactorily addresses the previous concerns of the reviewer.

**Reviewer #2 (Public review):**
SummaryLe Roy et al quantify wing morphology and wing kinematics across twenty eight and eight hoverfly species, respectively; the aim is to identify how weight support during hovering is ensured across body sizes. Wing shape and relative wing size vary non-trivially with body mass, but wing kinematics are reported to be size-invariant. On the basis of these results, it is concluded that weight support is achieved solely through size-specific variations in wing morphology, and that these changes enabled hoverflies to decrease in size. Adjusting wing morphology may be preferable compared to the alternative strategy of altering wing kinematics, because kinematics may be subject to stronger evolutionary and ecological constraints, dictated by the highly specialised flight and ecology of the hoverflies.StrengthsThe study deploys a vast array of challenging techniques, including flight experiments, morphometrics, phylogenetic analyses, and numerical simulations; it so illustrates both the power and beauty of an integrative approach to animal biomechanics. The question is well motivated, the methods appropriately designed, and the discussion elegantly places the results in broad biomechanical, ecological, and evolutionary context.

We thank the reviewer for appreciating the strengths of our study.

Weaknesses(1) In assessing evolutionary allometry, it is key to pinpoint the variation expected from changes in size alone. The null hypothesis for wing morphology is well-defined (isometry), but the equivalent predictions for kinematic parameters, although specified, are insufficiently justified, and directly contradict classic scaling theory. A detailed justification of the "kinematic similarity" assumption, or a change in the null hypothesis, would substantially strengthen the paper, and clarify its evolutionary implications.

We agree with the reviewer that a clearly articulated null hypothesis is crucial for interpreting scaling relationships. In fact, when carefully reviewing our manuscript, we realized that we nowhere did so, and which might have led to a misinterpretation of this. In the revised manuscript, we therefore now explicitly state our newly defined null hypotheses (lines 120–125, 340-352), and how we tested these (lines 359-360).

In fact, we define two alternative null hypotheses: (1) weight support is maintained across sizes using allometric scaling of wing morphology only, and thus wingbeat kinematics are kept constant (kinematic similarity); (2) weight support is maintained across sizes using allometric scaling of wingbeat kinematics, while wing morphology scales isometrically (morphological similarity).

According to the first null hypothesis, the second-moment-of-area of the wing should scale linearly with body mass, resulting in negative allometry of *S2* relative to body mass (S_2_∼m^1^ <m^4/3^). According to the second null hypothesis, the product of wingbeat frequency and amplitude should scale with mass under negative allometry (ω∼ƒ A_ϕ_∼m^-1/6^). We test these alternative null hypotheses using Phylogenetic Generalized Least Square (PGLS) regressions of the morphology and kinematics metrics against the body mass.

Furthermore, in our revised manuscript, we now also better explain the use of "kinematic similarity" assumption as a theoretical scenario, that is physically, biomechanically nor physiological sustainable across sizes, but that we merely use to define our null hypotheses (lines 340-351). This is made particularly explicit in a new subsection named “Theoretical considerations” (lines 448–461). Note that our second null hypothesis is thus not that hoverflies fly under "kinematic similarity", but that wingbeat kinematics scales under negative allometry (ω∼ƒ A_ϕ_∼m^-1/6^), which we assume is in line with the classic scaling theory that the reviewer refers to.

We sincerely thank the reviewer for making us aware that we did not explicitly state our null hypotheses, and that introducing these new null hypotheses removed the confusion about the assumptions in our study.

(2) By relating the aerodynamic output force to wing morphology and kinematics, it is concluded that smaller hoverflies will find it more challenging to support their body mass--a scaling argument that provides the framework for this work. This hypothesis appears to stand in direct contrast to classic scaling theory, where the gravitational force is thought to present a bigger challenge for larger animals, due to their disadvantageous surface-to-volume ratios. The same problem ought to occur in hoverflies, for wing kinematics must ultimately be the result of the energy injected by the flight engine: muscle. Much like in terrestrial animals, equivalent weight support in flying animals thus requires a positive allometry of muscle force output. In other words, if a large hoverfly is able to generate the wing kinematics that suffice to support body weight, an isometrically smaller hoverfly should be, too (but not vice versa). Clarifying the relation between the scaling of muscle mechanical input, wing kinematics, and weight support would help resolve the conflict between these two contrasting hypotheses, and considerably strengthen the biomechanical motivation and evolutionary interpretation.

We agree with the reviewer that, due to disadvantageous surface-to-volume ratios, larger animals are more challenged to maintain weightsupport, and that this is also the case for hovering hoverflies. In the current manuscript, we do not aim to challenge this universal scaling law of muscle force with body mass.

Instead, we here focus merely on how the flight propulsion system (wing morphology and kinematics) scale with size, and how this allows hovering hoverflies to maintain weight support. We also fully agree with the reviewer that in theory, “if a large hoverfly is able to generate the wing kinematics that suffice to support body weight, an isometrically smaller hoverfly should be, too”. This aligns in fact with our second null hypothesis where wingbeat frequency should scale as ƒ∼m^-1/6^, to maintain weight support under morphological isometry.

In our study, we show that this null hypothesis is rejected (lines 511-517, and line 525), and thus hoverflies primarily adjust their wing morphology to maintain in-hovering weight support across sizes, and wingbeat kinematics is in fact highly conserved. Why this specific flight kinematics is so strongly conserved is not known, and thus a key topic in the discussion section of our manuscript.

We agree with the reviewer that muscle physiology might be an important driver for this conserved kinematics, but also aerodynamic efficiency and maneuverability could be key aspects here. In our revised manuscript, we now discuss these three aspects in more detail (lines 762-775). Also, we here now also mention that we aim to address this outstanding question in future studies, by including muscle physiology in our animal flight studies, and by studying the aerodynamics and maneuver kinematic of hoverflies in more detail.

Moreover, in our revised introduction section, we now also mention explicitly that the capability for maintaining in-flight weight support scales inversely with animal size, due to the negative isometric scaling of muscle force with body mass (line 52-56). Furthermore, we removed all statements that might suggest the opposite. We hope that these adjustments helped resolve the apparent conflict between our null hypotheses and general muscle scaling laws.

Finally, in the Discussion section (lines 770-775), we now more explicitly acknowledge that wing motion is ultimately driven by the flight motor musculature, and that a full biomechanical interpretation must consider the scaling of muscle mechanical input alongside wing kinematics and morphology. While we decided to keep the focus primarily on aerodynamic constraints in this study, we agree that future work integrating both aerodynamic and physiological scaling will be essential to fully resolve these contrasting perspectives.

(3) One main conclusion-- that miniaturization is enabled by changes in wing morphology--is insufficiently supported by the evidence. Is it miniaturization or "gigantism" that is enabled by (or drives) the non-trivial changes in wing morphology? To clarify this question, the isolated treatment of constraints on the musculoskeletal system vs the "flapping-wing based propulsion" system needs to be replaced by an integrated analysis: the propulsion of the wings, is, after all, due to muscle action. Revisiting the scaling predictions by assessing what the engine (muscle) can impart onto the system (wings) will clarify whether non-trivial adaptations in wing shape or kinematics are necessary for smaller or larger hovering insects (if at all!).In many ways, this work provides a blueprint for work in evolutionary biomechanics; the breadth of both the methods and the discussion reflects outstanding scholarship.

In response to the first review round, we have removed all references to “miniaturization,” as our data does not allow us to infer evolutionary trajectories of body size (i.e., whether lineages have become smaller or larger over time). We now frame our conclusion more conservatively: that changes in wing morphology enable small hoverflies to maintain weight support despite the aerodynamic disadvantages imposed by isometric scaling.

We fully agree that an integrated biomechanical framework, explicitly linking muscle mechanical output with wing kinematics and morphology, would significantly strengthen the study. However, we believe that performing an integrated analysis assessing the scaling of muscle input into the wing is beyond the current scope, which focuses specifically on the aerodynamic consequences of morphological and kinematic variation (see reply above).

**Reviewer #3 (Public review):**
This paper addresses an important question about how changes in wing morphology vs. wing kinematics change with body size across an important group of high-performance insects, the hoverflies. The biomechanics and morphology convincingly support the conclusions that there is no significant correlation between wing kinematics and size across the eight specific species analyzed in depth and that instead wing morphology changes allometrically. The morphological analysis is enhanced with phylogenetically appropriate tests across a larger data set incorporating museum specimens.The authors have made very extensive revisions that have significantly improved the manuscript and brought the strength of conclusions in line with the excellent data. Most significantly, they have expanded their morphological analysis to include museum specimens and removed the conclusions about evolutionary drivers of miniaturization. As a result, the conclusion about morphological changes scaling with body size rather than kinematic properties is strongly supported and very nicely presented with a strong complementary set of data. I only have minor textual edits for them to consider.

We thank the reviewer for this positive feedback. We are pleased to hear that the revised manuscript is satisfactory.

**Reviewer #2 (Recommendations For The Authors):**
My main remaining qualm remains the null hypothesis for the scaling of kinematic parameters - all weaknesses come back to this point. I appreciate that the authors now specify an expectation, but they offer no justification. This is a problem, because the expectation dictates the interpretation of the results and is thus crucial to some of the key claims (including one in the paper title!): the choice made by the authors indeed implies that hovering is harder for small hoverflies, so that the reported changes in size-specific wing morphology are to be interpreted as an adaptation that enables miniaturization. However, why is this choice appropriate over alternatives that would predict the exact opposite, namely that hovering is harder for larger hoverflies?In my original review, I suggested that the authors may address this key question by considering the scaling of muscle mechanical output, and provided a quick sketch of what such an argument would look like, both in classic textbook scaling theory, and in the framework of more recent alternative approaches. The authors have decided against an implementation of this suggestion, providing various version of the following justification in their reply: "our study focuses precisely on this constraint on the wing-based propulsion system, and not on the muscular motor system." I am puzzled by this distinction, which also appears in the paper: muscle is the engine responsible for wing propulsion. How can one be assessed independent of the other? The fact that the two must be linked goes straight to the heart of the difficulty in determining the null hypotheses for the allometry of kinematic and dynamic parameters: they must come from assertions on how muscle mechanical output is expected to vary with size, and so couple muscle mechanical output to the geometry of the wing-based propulsion system. What if not muscle output dictates wing kinematics?I fully agree with the authors that null hypotheses on kinematic parameters are debatable. But then the authors should debate their choice, and at least assess the plausibility of its implications (note that the idea of "similarity" in scaling does not translate to equal or invariant, but is tied closely to dimensional analysis - so one cannot just proclaim that kinematic similarity implies no change in kinematic parameters). I briefly return to the same line of argument I laid out in the initial review to provide such an assessment:Conservation of energy implies:W = 1/2 I ω2where I is the mass moment of inertia and W is the muscle work output. Under isometry, I ∝m5/3, the authors posit ω ∝m0, and it follows at once that they predict W ∝m5/3. That is, the "kinematic similarity" hypothesis presented in the paper implies that larger animals can do substantially more work per unit body mass than small animals (unless the author have an argument why wing angular velocity is independent of muscle work capacity, and I cannot think of one). This increase in work output is in contradiction with the textbook prediction, going all the way back to Borelli and Hill: isogeometric and isophysiological animals ought to have a constant mass-specific work output. So why, according to the authors, is this an incorrect expectation, ie how do they justify the assumption ω ∝m0 and its implication W ∝m5/3? How can larger animals do more mass-specific work, or, equivalently, what stops smaller animals from delivering the same mass-specific work? If non-trivial adaptations such as larger relative muscle mass enable larger animals to do more work, how does this fit within the interpretation suggested by the authors that the aerodynamics of hovering require changes in small animals?A justification of the kinematic similarity hypothesis, alongside answers to the above questions, is necessary, not only to establish a relation to classic scaling theory, but also because a key claim of the paper hinges on the assumed scaling relationship: that changes in wing morphology enable hovering in small hoverflies. If I were to believe Borelli, Hill and virtually all biomechanics textbooks, the opposite should be the case: combing constant mass-specific work output with eq. 1, one retrieves F∝m2/3, so that weight support presents a bigger challenge for larger animals; the allometry of wing morphology should then be seen as an adaptation that enables hovering in larger hoverflies - the exact opposite of the interpretation offered by the authors.Now, as it so happens, I disagree with classic scaling theory on this point, and instead believe that there are good reasons to assume that muscle work output varies non-trivially with size. The authors can find a summary of the argument for this disagreement in the initial review, or in any of the following references:Labonte, D. A theory of physiological similarity for muscle-driven motion. PNAS, 2023, 120, e2221217120Labonte, D.; Bishop, P.; Dick, T. & Clemente, C. J. Dynamics similarity and the peculiar allometry of maximum running speed. Nat Comms., 2024, 15, 2181Labonte, D. & Holt, N. Beyond power limits: the kinetic energy capacity of skeletal muscle. J Exp Bio, 2024, 227, jeb247150Polet, D. & Labonte, D. Optimal gearing of musculoskeletal systems. Integr Org Biol, 2024, 64, 987-10062024I am asking neither that the authors agree with the above references nor that they cite them. But I do expect that they critically discuss and justify their definition of kinematic similarity, its relation to expectation from classic scaling theory, and the implications for their claim that hovering is harder for small animals. I do note that the notion of "physiological similarity" introduced in the above references predicts a size-invariant angular velocity for small animals, that small animals should be able to do less mass-specific work, and that average muscle force output can grow with positive allometry even for isogeometric systems. These predictions appear to be consistent with the data presented by the authors.

We agree with the reviewer that our null hypothesis was not clearly articulated in our previous version of the manuscript, and that this might have led to a misinterpretation of the merits and limitations of our study. In the revised manuscript, we therefore now explicitly introduce our null hypotheses in the Introduction (lines 120–125), we define these in the Methods section (lines 340–360), test these in the Results section (lines 511–517), and reflect on the results in the Discussion (lines 602–610). We thank the reviewer for pointing out this unclarity in our manuscript, because revising it clarified the study significantly. See our replies in the “Public Review” section for details.

Minor pointsL56: This is somewhat incomplete and simplistic; to just give one alternative option, weight support with equivalent muscle effort could also be ensured by a change in gearing (see eg Biewener's work). It is doubtful whether weight support is a strong selective force, as any animal that can move will be able to support its weight. The impact of scaling on dynamics is thus arguably more relevant.

We thank the reviewer for pointing out that our original sentence may be too simplistic. We now briefly mention alternative mechanisms (suggested by the reviewer) to provide more nuance (line 56-58).

L58: I am not aware of any evidence that smaller animals have reduced the musculature dedicated to locomotion beyond what is expected from isometry; please provide a reference for this claim or remove it.

We removed that claim.

The authors use both isometry and geometric similarity. As they also talk about muscle, solely geometric similarity (or isogeometry) may be preferable, to avoid confusion with isometric muscle contractions.

To avoid confusion, we now use “geometric similarity” wherever the use of isometry might be ambiguous.

L86: negative allometry only makes sense if there is a justified expectation for isometry - I suggest to change to "The assumed increase in wingbeat frequency in smaller animals" or similar, or to clarify the kinematic similarity hypothesis.

We edited the sentence as suggested.

L320: This assertion is somewhat misleading. Musculoskeletal systems are unlikely to be selected for static weight support. Instead, they need to allow movement. Where movement is possible, weight support is trivially possible, and so weight support should rarely, if ever, be a relevant constraint. At most, the negative consequence of isometry on weight support would be that a larger fraction of the muscle mass needs to be active in larger animals to support the weight.

We fully agree with the reviewer that musculoskeletal systems are unlikely not selected for static loads, as the ability to move dynamically in the real world is crucial for survival. That said, we here look at hovering flight, which is far from static. In fact, hovering flight is among the energetic most costly movement patterns found in nature, due to the required high-frequency wingbeat motions (Dudley 2002). Rapid maneuvers are of course more power demanding, but hovering is a good proxy for this. For example, in fruit flies maximum force production in rapid evasive maneuvers are only two times the force produced during hovering (Muijres et al., 2014).

We agree with the reviewer that it is important to explicitly mention the differences in functional demands on the motor system in hovering and maneuvering flight, and thus we now do so in both the introduction and discussion sections (lines 116-118 and 762-765, respectively).

Dudley, Robert. The biomechanics of insect flight: form, function, evolution. Princeton university press, 2002.Muijres, F. T., et al. "Flies evade looming targets by executing rapid visually directed banked turns." Science 344.6180 (2014): 172-177.

**Reviewer #3 (Recommendations For The Authors):**
Throughout, check use of "constrains" vs. "constraints"

Thank you for pointing this out. We have corrected these errors.

Line 52 do you mean lift instead of thrust?

We agree with the reviewer that the use of “thrust” might be confusing in the context of hovering flight, and thus we replaced “flapping-wing-based aerodynamic thrust-producing system” with the “flapping-wing-based propulsion system”. This way, we no longer use the word thrust in this context, and only use lift as the upward-directed force required for weight support.

Line 60 "face also constrains" wording

Corrected.

Line 79 Viscous forces only "dominate" at Re<1 and so this statement only refers to very very small insects which I suspect are far below the scale of the hoverflies considered (likely Re ~100) although maybe not for the smallest 3 mg ones?

Indeed, viscous forces do not “dominate” force production at the Reynolds numbers of our flying insects. We thank the reviewer for pointing out this incorrect statement, which we corrected in the revised manuscript.

Line 85 again thrust doesn't seem to be right

Agreed. See reply 3.2.

533 "maximized" should probably be "increased"

We now use “increased”.

Line 705-710 The new study by Darveau might help resolve this a bit because of the reliability of this relationship across and between orders. Darveau, C.-A. (2024). Insect Flight Energetics And the Evolution of Size, Form, And Function. Integrative And Comparative Biology icae028.

We thank the reviewer for this highly relevant reference, which was unfortunately not included in the original manuscript. In connection with this work, we now further discuss the relationship between wing size allometry and deviations from the expected scaling of wingbeat frequency (lines 730-735).